# Alternative splicing expands the antiviral IFITM repertoire in Chinese rufous horseshoe bats

Nelly S. C. Mak[1,2], Jingyan Liu[3], Dan Zhang[3], Jordan Taylor[1,2], Xiaomeng Li[3], Kazi Rahman[4], Feiyu Chen[2,3], Siddhartha A. K. Datta[4], Kin Kui Lai[4], Zhengli Shi[5], Nigel Temperton[6], Aaron T. Irving[7,8,9,10]*, Alex A. Compton[4]*, Richard D. Sloan[1,2,3]*

1 Centre for Inflammation Research, Institute for Regeneration and Repair, University of Edinburgh, Edinburgh, United Kingdom, 2 Deanery of Biomedical Sciences, Edinburgh Medical School, University of Edinburgh, Edinburgh, United Kingdom, 3 Zhejiang University-University of Edinburgh Institute, Zhejiang University School of Medicine, Zhejiang University, Haining, China, 4 HIV Dynamics and Replication Program, National Cancer Institute, National Institutes of Health, Frederick, Maryland, United States of America, 5 CAS Key Laboratory of Special Pathogens and Biosafety, Wuhan Institute of Virology, Chinese Academy of Sciences, Wuhan, China, 6 Viral Pseudotype Unit, Medway School of Pharmacy, University of Kent and Greenwich Chatham Maritime, Kent, United Kingdom, 7 Second Affiliated Hospital, School of Medicine, Zhejiang University, Hangzhou, China, 8 Centre for Infection, Immunity & Cancer, Zhejiang University-University of Edinburgh Institute, Zhejiang University School of Medicine, Haining, China, 9 College of Medicine and Veterinary Medicine, University of Edinburgh, Edinburgh, United Kingdom, 10 Biomedical and Health Translational Research Centre of Zhejiang Province, Haining, China

* aaronirving@intl.zju.edu.cn (ATI); alex.compton@nih.gov (AAC); richard.sloan@ed.ac.uk (RDS)

**Data Availability Statement:** All relevant data are within the paper and its supporting data files. Code scripts for the retrieval and analysis of published

## Abstract

Species-specific interferon responses are shaped by the virus-host arms race. The human interferon-induced transmembrane protein (IFITM) family consists of three antiviral *IFITM* genes that arose by gene duplication. These genes restrict virus entry and are key players in antiviral interferon responses. The unique IFITM repertoires in different species influence their resistance to viral infections, but the role of IFITMs in shaping the enhanced antiviral immunity of reservoir bat species is unclear. Here, we identified an *IFITM* gene in Chinese rufous horseshoe bat, a natural host of severe acute respiratory syndrome (SARS)-related coronaviruses, that is alternatively spliced to produce two IFITM isoforms in native cells as shown by transcriptomics. These bat IFITMs have conserved structures *in vitro* as demonstrated by circular dichroism spectroscopy, yet they exhibit distinct antiviral specificities against influenza A virus, Nipah virus and coronaviruses including SARS-CoV, SARS-CoV-2 and MERS-CoV. In parallel with human IFITM1-3, bat IFITM isoforms localize to distinct sites of virus entry which influences their antiviral potency. Further bioinformatic analysis of IFITM repertoires in 206 mammals reveals that alternative splicing is a recurring strategy for IFITM diversification, albeit less widely adopted than gene duplication. These findings demonstrate that alternative splicing is a key strategy for evolutionary diversification in the IFITM family. Our study also highlights an example of convergent evolution where species-specific selection pressures led to expansion of the IFITM family through multiple means, underscoring the importance of IFITM diversity as a component of innate immunity.

genomic and transcriptomic datasets are deposited at https://github.com/nellymak1/Mammalian_ IFITM_splicing_analysis.

**Funding:** NM is funded by a joint Ph.D. studentship from the University of Edinburgh and Leiden University Medical Center. JT is funded by a Medical Research Council doctoral training program (MR/N013166/1) https://www.ukri.org/ councils/mrc/. ATI is funded by a Key grant from the National Science Foundation of Zhejiang Province (Z23C010003) and a National Science Foundation Research Fund for International Excellent Young Scientists (RFIS-II, 82350610279) https://www.nsfc.gov.cn. The funders had no role in the study design, data collection and analysis, decision to publish, or preparation of the manuscript.

**Competing interests:** The authors have declared that no competing interests exist.

## Author summary

Zoonotic transmission occurs when viruses 'jump' from animals into human. This may lead to viral outbreaks such as the COVID-19 pandemic, posing a significant threat to public health. Bats are the origin of many zoonotic viruses as their unique immunity may allow them to carry viruses without developing disease. Interferon-induced transmembrane proteins (IFITMs) are important antiviral proteins that have been shown to influence the pathogenesis of viral infections. It is currently unclear whether IFITMs also contribute to the high viral tolerance of bats, so characterization of bat IFITMs is needed to identify factors that predispose species to act as viral reservoirs. Here, we find that the Chinese rufous horseshoe bat, a natural host of SARS-related coronaviruses, adopts a distinct strategy known as alternative splicing to functionally diversify their IFITM family. We also demonstrate that alternative splicing is a recurring strategy in the evolution of IFITMs and is evident in at least 75 mammalian species. Our study therefore provides novel insights into how epidemiologically significant species could take advantage of different evolutionary strategies to enhance their resistance to viruses.

## Introduction

Human and other animals possess interferon-stimulated genes (ISGs) that encode antiviral proteins which, in part, dictate the permissiveness of cells to virus infections. Interferon-induced transmembrane proteins (IFITMs) are a family of antiviral proteins that inhibit virus entry into target cells and are upregulated by type I, II and III interferons (IFN) [1]. The human IFITM family consists of three antiviral IFITMs with high sequence similarity (IFITM1, IFITM2 and IFITM3) and two IFITMs that are not interferon-inducible and not known to be involved in immunity (IFITM5 and IFITM10) [2]. IFITM3 is the best-studied owing to its potency against Influenza A virus (IAV) and many other viruses, such as HIV-1 and dengue virus [3,4]. The effect of IFITMs on coronaviruses (CoVs) is less clear cut. While CoVs are generally inhibited by overexpression of IFITMs, endogenous IFITMs have little effect, or may even promote SARS-CoV and SARS-CoV-2 infection [5–7]. An exception is human coronavirus OC43, which is always enhanced by IFITM2 and IFITM3 [8,9]. Beyond their antiviral activity, IFITMs have pleiotropic effects such as regulating interferon production, adaptive T- and B-cell responses, and influencing cancer growth [10,11].

IFITMs inhibit virus-cell membrane fusion by mechanisms that are not fully understood. The best current working model suggests that the IFITM3 amphipathic helix inserts into membranes to induce a negative membrane curvature that disfavors the formation of a fusion pore, a critical step in membrane fusion [12–15]. Cholesterol binding was recently shown to be crucial for IFITM3 antiviral activity [16,17]. However, the role played by cholesterol in IFITM-mediated inhibition of virus entry is less straightforward with several alternative mechanisms being proposed [18–21]. IFITMs contain a canonical CD225 domain that is composed of an intramembrane domain and a conserved intracellular loop, and it contains residues that can be post-translationally modified to alter IFITM function [22–24]. The $_{20}$YXXΦ$_{23}$ motif in the N-terminal domain of IFITM2 and IFITM3 serves as an endocytic signal for their localization to endolysosome membranes, whereas the absence of this motif in IFITM1 results in its surface localization [25]. In addition to altering host cell membranes, IFITMs are incorporated into nascent virions and can reduce their infectivity [26–28]. This antiviral property termed

"negative imprinting" likely occurs via changes in virus membrane properties, but may also involve direct interaction of IFITMs with viral envelope proteins [29].

A problem faced by the ISG biology field is our limited understanding of non-human ISGs. Characterizing ISG functions in reservoir species is however an emerging area with immense public health importance, as it may yet uncover mechanisms that enable these species to harbor zoonotic viruses with the potential to spillover into humans. For instance, it was demonstrated that the higher expression of IFITMs in ducks compared to chicken underlies their tolerance to various strains of avian influenza viruses [30]. Among reservoir species, bats have perhaps attracted the most interest in recent years. Bats make up 21% of all mammalian species and are the only mammals utilizing powered flight, they likely harbor more zoonotic viruses compared to other mammals and may carry them over long distances [31–33]. Examples of bat-originated viruses that have caused outbreaks in the human population are Marburg virus, Nipah virus, and coronaviruses including Middle East respiratory syndrome CoV (MERS-CoV), SARS-related CoVs and the common cold human coronavirus 229E (HCoV-229E) [34–36]. SARS-related CoVs likely originated from horseshoe bats, with viruses closely related to SARS-CoV and SARS-CoV-2 being detected in bats in the *Rhinolophidae* family [37–39]. Immune adaptations in bats predispose them to acting as viral reservoirs, and constitutive expression of interferons and ISGs that may contribute to their high viral tolerance has been observed in several species [40–45]. Benfield *et al.* reported that microbat IFITM3 inhibits IAV and lyssaviruses, this was however shown with only one bat IFITM isoform in human cells [46,47]. There is therefore a need to characterize the function of IFITMs in the native context of other reservoir bat species to bridge the knowledge gap.

The strong selection pressure imposed by viruses on the host immune system shapes the evolution of species-specific ISG repertoires [48]. In fact, ISGs have a higher rate of gene duplication compared to other genes [44,49]. In the case of IFITMs, species-specific gene duplication generates distinct IFITM families [2,50]. In addition to gene duplication, the formation of novel transcripts is a significant evolutionary mechanism for the generation of proteomic diversity [51]. Both mechanisms can increase the available mutational space for the generation and positive selection of novel functions. Alternative splicing of human IFITMs alters their function: an N-terminus truncated IFITM2 isoform specific to immune cells displays altered antiviral specificity against HIV-1 strains [52]; while the IFITM3 rs12252-C allele is predicted to produce an aberrantly spliced N-terminus truncated IFITM3 which is associated with severe influenza, HIV-1 and COVID-19, although attempts to identify this mutant protein have thus far been unsuccessful [53–58]. Novel transcripts and gene duplication also lead to the accumulation of species- or lineage-specific mutations, which is evident among ISGs [59]. For instance, evolutionary analysis of IFITMs revealed strong positive selection on the branch leading to bat IFITMs and identified a highly variable residue within IFITMs [47]. Together, these evolutionary strategies result in unique ISG repertoires with different antiviral capacities across species. While phylogenetic studies have examined *IFITM* genes in non-human species, the potential role of alternative splicing underscores the importance of studying these IFITM repertoires at a transcript level, which has not been done [47,60,61].

In this study, we use a combination of biochemical and molecular tools to characterize the expression, structure and antiviral function of IFITMs in an epidemiologically significant bat species. We show that *Rhinolophus sinicus*, a natural host of SARS-related CoVs, uses alternative splicing to generate IFITM functional diversity which could contribute to their high viral tolerance. We then examine IFITM transcripts in 206 mammals more broadly using bioinformatics tools to gain insights into how alternative splicing contributes to IFITM diversity across species.

## Results

### *R. sinicus* possesses an *IFITM* gene that encodes two IFITM splice variants

To characterize the IFITM repertoire in *R. sinicus*, we searched the available *R. sinicus* genome on NCBI for homologs of human immune-related *IFITM1-3*. We identified a single gene, *LOC109436297*, that shows highest homology with *IFITM3* and is flanked by *B4GALNT4*, similar to the human *IFITM* locus (**Fig 1A**). Alternative first exon splicing of the gene generates two predicted protein-coding mRNA transcripts that are distinct at the N-terminus (XM_019714804.1 and XM_019714805.1). *R. sinicus* thus potentially expresses two IFITM isoforms which we refer to as rsIFITMa (XP_019570363.1) and rsIFITMb (XP_019570364.1), and collectively as rsIFITMs in this article. Orthologs of *IFITM5* and *IFITM10* are also present in the *R. sinicus* genome but they do not contain interferon-stimulated response elements (ISRE) around the transcription start site and their human orthologs are not known to be antiviral, so they were not examined in this study (**S1A Fig**).

Pairwise amino acid sequence alignment of rsIFITMa and rsIFITMb indicates a 79.4% sequence identity, with most variation at the N-terminus (**S1B Fig**). rsIFITMa contains the $_{20}$YEML$_{23}$ endocytic motif, while rsIFITMb resembles IFITM1 in that it has a truncated N-terminus lacking this motif. Alignment of human and *R. sinicus* IFITMs shows that 45 out of 50 residues in the canonical CD225 domain are conserved. Importantly, residues that undergo post-translational modifications such as ubiquitination (at Lys83, Lys88 and Lys104) and palmitoylation (at Cys71, Cys72 and Cys105) are conserved across these proteins, in line with a previous study that showed conservation of these residues in bat IFITMs [22,47]. However, Lys24, which was reported to be the most robustly ubiquitinated in IFITM3 is lost in rsIFITMb. Residues involved in IFITM oligomerization (Gly91 and Gly95) are also conserved [9,62].

To examine whether rsIFITMa and rsIFITMb are natively expressed, we performed RNA sequencing on *R. sinicus* kidney epithelial (RsKT.01) cells [63]. Both rsIFITM isoforms were endogenously expressed and upregulated upon poly(I:C) treatment or infection with vesicular stomatitis virus (VSV), which elicits an interferon response (**Fig 1B–1D**). Similarly, RT-qPCR shows the upregulation of rsIFITMa and rsIFITMb in response to poly(I:C) stimulation (**S2A Fig**). Western blotting confirms the expression of rsIFITM at the protein level, with two bands likely representing the two isoforms with different molecular weights (**S2B–S2C Fig**). To confirm that the upper and lower bands observed on the western blots correspond to rsIFITMa and rsIFITMb respectively, we attempted to abolish their expression by CRISPR/Cas9-mediated knockout. Targeting the first exon of *rsIFITMa* led to the loss of the upper band, indicating that it represents rsIFITMa expression (**S2D Fig**). These findings suggest that *R. sinicus* possesses an *rsIFITM* gene that encodes two IFITM splice variants with distinct N-terminal domains.

### *R. sinicus* IFITMs have a structurally and functionally conserved amphipathic helix

The human IFITM3 amphipathic helix is required for antiviral activity by binding cholesterol and increasing membrane order [12,16]. The rsIFITMa and rsIFITMb isoforms share an identical amphipathic helix, which only differs from the human IFITM amphipathic helices at the last residue (**Fig 2A**). *In silico* analyses show that this amino acid substitution preserves the helical structure of the rsIFITM amphipathic helix, while slightly reducing its mean hydrophobic moment and increasing its hydrophobicity, implying a reduced amphipathicity compared to that of human IFITM2 or IFITM3 (IFITM2/3) (**S3A Fig**). Wider analysis of mammalian IFITMs reveals that bat IFITMs have amphipathic helices that are less amphipathic compared

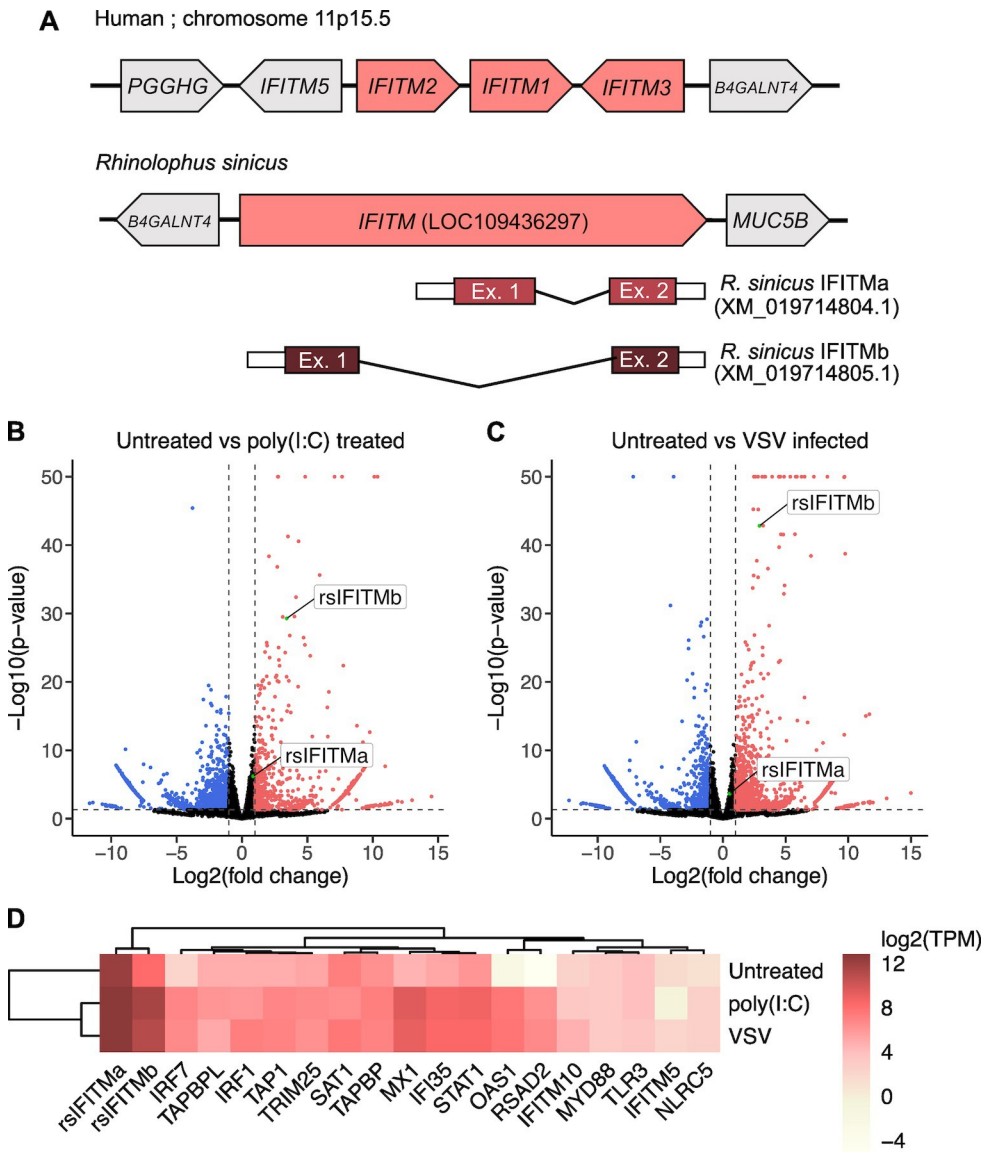

**Fig 1.** *Rhinolophus sinicus* **possess an *IFITM* gene that encodes two distinct IFITM isoforms.** A. Schematic representation of the human *IFITM1-3* loci and *R. sinicus IFITM* (*LOC109436297*) with flanking genes. The *R. sinicus IFITM* gene was identified by BLAST and produces two transcripts, IFITMa (XM_019714804.1) and IFITMb (XM_019714805.1). **B-C.** Volcano plots of differential expression of gene transcripts comparing untreated cells versus poly(I:C)-treated (B) or VSV-infected (C) cells. Upregulated (red) and downregulated (blue) gene transcripts are indicated. Data points with -log10(p-value) above 50 are plotted at the y-axis upper limit. **D.** Heatmap showing normalized expression log2(TPM) of rsIFITM transcripts and selected ISGs for each condition. Gene-level TPMs were calculated as the sum of transcript-level TPMs for genes excluding *rsIFITM*. TPM; transcripts per million.

to human and other mammals (**S3B Fig**). Circular dichroism spectroscopy of synthetic peptides confirmed similar and substantial helical content (56.0–63.1%) between human and *R. sinicus* IFITM amphipathic helices (**Fig 2B–2C**). Next, we tested whether the rsIFITM amphipathic helix retained the ability to bind cholesterol using a previously established fluorescence-based *in vitro* binding assay [16]. Peptide binding to the cholesterol analog NBD-cholesterol was measured by fluorescence intensity and polarization. All human and *R. sinicus* IFITM amphipathic helices exhibited cholesterol binding activity, with the rsIFITM amphipathic

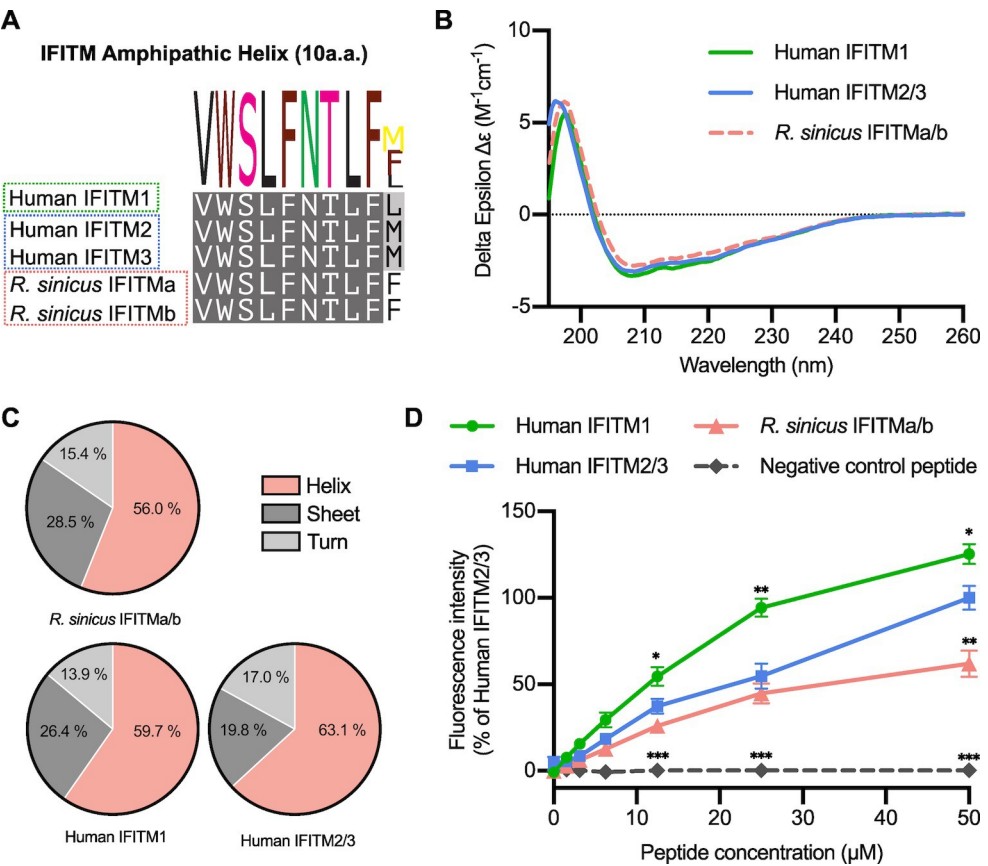

**Fig 2. The amphipathic helix of *R. sinicus* IFITMs has conserved structure and function. A.** Protein sequence alignment of the amphipathic helix of immune-related human and *R. sinicus* IFITMs, with the consensus sequence shown as a sequence logo above. **B-C.** Structures of IFITM amphipathic helix peptides were characterized by circular dichroism spectroscopy to determine their secondary structure compositions. Spectra in (B) represent the average of six acquisitions. **D.** NBD-cholesterol fluorescence intensity was measured following incubation of IFITM amphipathic helix peptides (0–50 μM) with NBD-cholesterol (500 nM). Data points are normalized to 50 μM human IFITM2/3. Error bars represent SEM of averages from 3 independent experiments. Statistical significance of difference between human IFITM2/3 and another peptide was determined by one-way ANOVA with Dunnett's test; *p<0.05, **p<0.01, ***p<0.001.

helix binding cholesterol to a similar extent as the human IFITM2/3 amphipathic helix (**Figs 2D** and **S3C**).

To confirm that the rsIFITM amphipathic helix is sufficient to mediate IFITM antiviral activity, chimeric constructs were generated by substituting the amphipathic helix of human IFITM3 with that of rsIFITMs (IFITM3-AH [*R. sinicus*]) (**S3D Fig**). Expression of IFITM3 or IFITM3-AH [*R. sinicus*] led to potent inhibition of IAV infection, with no significant difference between the extent of inhibition (**S3E–S3F Fig**). Taken together, we show that the rsIFITM amphipathic helix has conserved secondary structure and function, albeit containing a mutation in the last amino acid.

### *R. sinicus* IFITM isoforms exhibit differential antiviral activity

To examine the antiviral activity of rsIFITMs, we expressed FLAG-tagged IFITM constructs in HEK293T cells and challenged them with IAV. Both rsIFITMs were well-expressed upon transfection at levels comparable to that observed following interferon induction in RsKT.01

cells (**Figs 3A and S2A**). To mitigate variation arising from inconsistent transfection efficiencies, transfected cells were immunostained and gated by flow cytometry to select only FLAG-positive cells for downstream analysis. Expression of human IFITM3 led to the strongest inhibition of IAV infection by 14-fold (**Figs 3B and S4B–S4C**). Inhibition of IAV infection by human IFITM1 and IFITM2 were significant but less pronounced, which is consistent with previous studies [3]. The antiviral potency of rsIFITMa was comparable to human IFITM3 and significantly greater than that of rsIFITMb (24-fold vs 3-fold inhibition, p = 0.002). The distinct antiviral potency between rsIFITMa and rsIFITMb was maintained in single-cycle infections at higher MOIs and in multi-cycle infections with IAV (**S5A–S5C and S6A–S6B Figs**). Inhibition of IAV by rsIFITMb was however not observed in stably transduced cells, indicating its modest antiviral potency relative to rsIFITMa. In addition, as an established mechanism of IFITM-mediated restriction, negative imprinting was also assessed [26,28]. Indeed, negative imprinting by rsIFITMa was evidenced by the reduced infectivity of IAV pseudotypes produced from rsIFITMa-expressing cells (**S5D Fig**).

To further test whether the two rsIFITM isoforms also restrict coronaviruses to different extents, lentiviral-based pseudotyped viral particles expressing spike protein from SARS-CoV, MERS-CoV or HCoV-229E were used. Expression of human IFITM1-3 inhibited the entry of all three CoV spike pseudotypes in line with previous reports (**Fig 3C–3E**) [20,64]. Both rsIFITM isoforms inhibited the CoV pseudotypes, and again, rsIFITMa showed a stronger restriction than rsIFITMb. rsIFITM-mediated restriction was also maintained at lower transfection dosages (**S7A–S7B Fig**). IFITM-mediated inhibition of HCoV-229E infection was confirmed with replication-competent HCoV-229E-GFP, which shows a similar pattern of inhibition by rsIFITMa and rsIFITMb (**Figs 3F and S4A and S4C**). The effect of IFITMs on SARS-CoV-2 infection *in vitro* largely depends on the experimental system [6]. In our hands, rsIFITMa inhibited SARS-CoV-2 pseudotypes expressing the Omicron variant spike but had little effect on Wuhan and Delta variants (**Fig 3G**). In contrast, rsIFITMb markedly enhanced the entry of SARS-CoV-2 pseudotypes of all three variants, supporting the divergence of rsIFITMa and rsIFITMb function. Notably, the Omicron variant has an altered cell entry pathway compared to the preceding Wuhan and Delta variants, where it favors an endosomal entry pathway over cell surface entry [65,66].

Next, we sought to confirm that rsIFITMs are antiviral in their native cell background. Expression of both rsIFITM isoforms significantly inhibited HCoV-229E infection in RsKT.01 cells (**Fig 3H**). The antiviral potencies of rsIFITMs in RsKT.01 cells were comparable to that in HEK293T cells and exhibited the same pattern, with rsIFITMa showing stronger inhibition. These results indicate that while both rsIFITM isoforms are capable of restricting virus entry, they have differential antiviral specificities.

## Distinct cellular localization of *R. sinicus* IFITM isoforms contributes to their antiviral specificity

Differential antiviral activity of human IFITM1-3 can be, at least in part, explained by their distinct cellular localization [67]. We hypothesized that rsIFITMa and rsIFITMb likewise localize to different cellular compartments, thus influencing their ability to restrict viruses depending on their route of entry. Immunofluorescence microscopy confirmed the well-documented pattern of human IFITM1 predominating at the plasma membrane, and IFITM2/3 preferentially colocalizing with the late endosome marker CD63 (**Figs 4A and S8**). This localization pattern was mirrored by rsIFITM isoforms expressed in HEK293T cells–rsIFITMa in internal compartments and rsIFITMb on or near the cell surface. Surface localization was quantified by measuring the percentage of FLAG signal at the plasma membrane, confirming that the

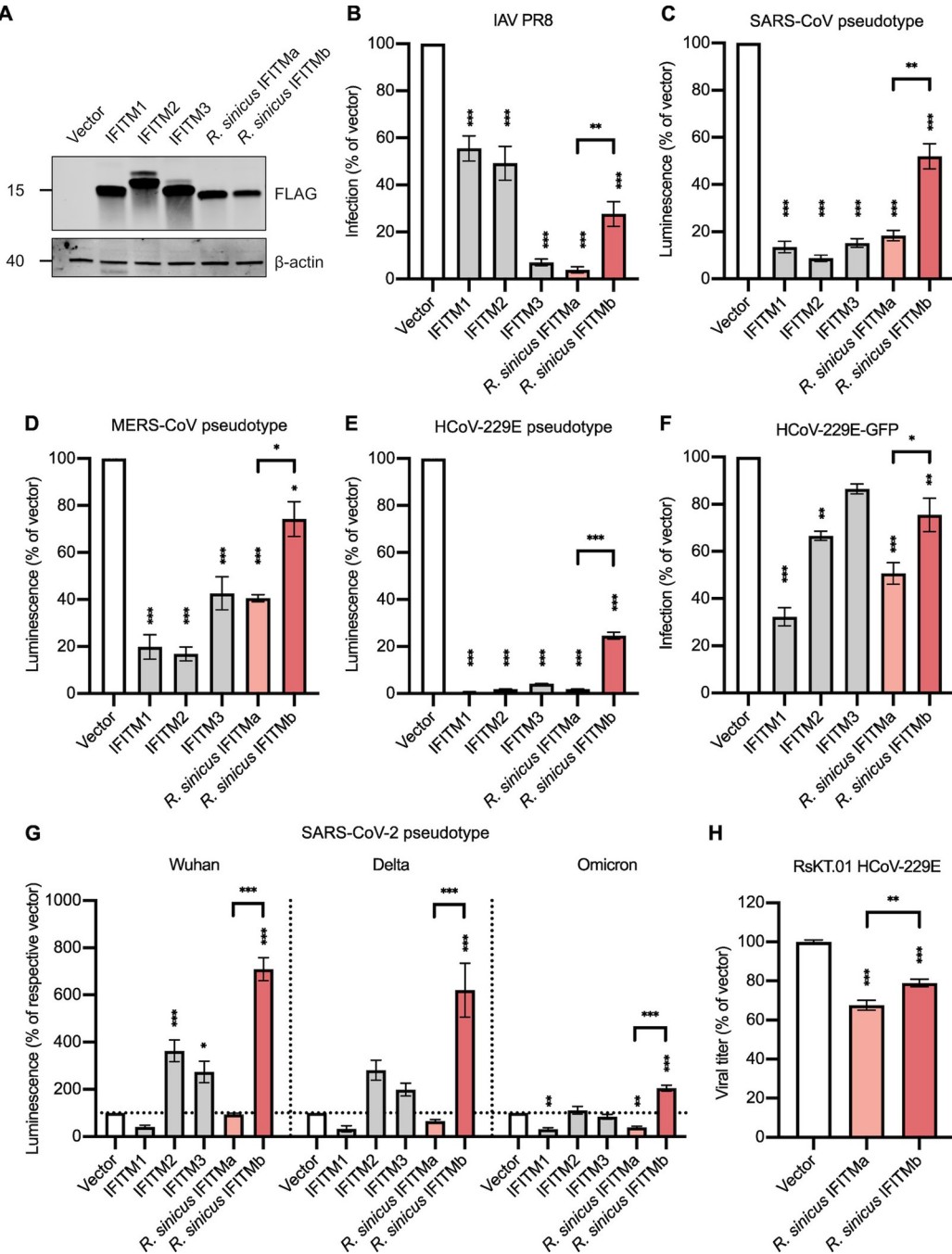

**Fig 3. *R. sinicus* IFITMs exhibit differential antiviral activity against IAV and coronaviruses. A.** HEK293T cells were transfected with the indicated FLAG-tagged IFITMs. IFITM expression was detected by western blotting at 24 hours post-transfection. **B.** Transfected HEK293T cells were infected with IAV at MOI = 0.05 and analyzed for NP-positive staining by flow cytometry at 18 hours post-infection. Error bars represent SEM of averages from 3 independent experiments, each performed in duplicate. **C-E.** HEK293T cells were co-transfected with the indicated IFITMs and coronavirus entry receptor (ACE2, DPP4 or APN) then transduced with SARS-CoV (C), MERS-CoV (D) or HCoV-229E (E) pseudotypes encoding a luciferase reporter. Cells were lysed and analyzed by luciferase assay after 48 hours. Error bars represent SEM of averages from 3 independent experiments, each performed in at least three replicates. **F.** Huh7.5 cells were transfected with the indicated IFITMs and infected with GFP-tagged HCoV-229E at MOI = 0.05. Cells were analyzed by flow cytometry at 18 hours post-infection. Error bars represent SEM of averages from 3 independent experiments, each performed in duplicate. **G.** HEK293T cells were co-transfected with the indicated IFITMs and ACE2 then transduced with SARS-CoV-2 pseudotypes expressing spike protein from Wuhan, Delta or Omicron variants and encoding a luciferase reporter. Cells were lysed and analyzed by luciferase assay after 48 hours. Error bars represent SEM of averages from 3

independent experiments, each performed in duplicate. **H.** RsKT.01 cells stably expressing the indicated IFITMs were infected with HCoV-229E at MOI = 0.5. Infectious virus titers in the supernatants were determined by TCID50 assay. Error bars represent SEM of averages from 3 independent experiments, each performed in duplicate. Statistical significance of difference between vector- and IFITM-expressing cells were determined by one-way ANOVA with Dunnett's test; statistical significance of difference between *R. sinicus* IFITMa- and IFITMb-expressing cells were determined by unpaired t-test; *$p < 0.05$, **$p < 0.01$, ***$p < 0.001$.

proportion of rsIFITMb found on the cell surface was significantly higher than that of rsI-FITMa (**Fig 4B**). Whereas the majority of rsIFITMa was found internally and colocalized more strongly with CD63 (**Fig 4C**).

We then examined whether the increased antiviral activity of rsIFITMa relative to rsI-FITMb could be explained by its accumulation in endosomes, which may be a more favored route of entry for the studied viruses. The N-terminal YXXΦ endocytic motif is required for the endosomal localization of IFITMs and phosphorylation of IFITM3 at Y20 prevents its endocytosis [25]. As a result, phosphomimetic mutation of this site (IFITM3 [Y20E]) causes constitutive localization at the cell surface [68–70]. We therefore tested the antiviral activity of N-terminal rsIFITMa and rsIFITMb mutants with altered localizations (**Figs 5A–5B and S9A–S9C**). Cell surface rsIFITMa [Y20E] was less able to inhibit HCoV-229E pseudotype entry compared to wild-type rsIFITMa (**Fig 5C**). On the other hand, the addition of a YXXΦ-containing N-terminus to rsIFITMb (rsIFITMb [Nt]) made it more antiviral, to a similar extent as wild-type rsIFITMa. Further incorporation of the Y20E mutation into rsIFITMb [Nt] partially removed its antiviral activity. We then hypothesized that the effect of localization on rsIFITM antiviral activity is dependent on the entry route of the virus. An attempt to redirect HCoV-229E spike cleavage to the plasma membrane using a cathepsin L inhibitor did not render rsIFITMb more antiviral (**S9D–S9E Fig**). This could be due to the endocytosis of HCoV-229E pseudotypes prior to fusion regardless of their spike cleavage status as proposed for other coronaviruses [71]. We therefore tested the ability of rsIFITMs to inhibit Nipah virus pseudotypes, as Nipah virus is a bat-borne paramyxovirus that predominantly enters target cells via pH-independent membrane fusion at the cell surface [72–74]. Contrary to previously tested viruses, stably expressed rsIFITMb inhibited Nipah pseudotype entry to a greater extent than rsIFITMa in HEK293T cells, consistent with the cell surface localisation of IFITMb (**Fig 5D**) [75]. These results suggest that the differential localization of rsIFITMs contributes to their distinct antiviral specificity, which is determined by the route of virus entry.

## Evolutionary convergence of IFITM diversification strategy

Independent evolution of the IFITM family in different species has led to distinct IFITM repertoires. To understand the evolutionary relationship between IFITMs in different species, we examined the phylogeny of IFITMs from commonly studied mammals and bats of epidemiological significance. Our analysis shows that mammalian IFITMs are grouped in two ways: IFITM isoform-specific clustering and species-specific clustering (**Fig 6**). Since IFITMs found in most species are not direct orthologs of human IFITMs according to their phylogenetic grouping, they were named IFITM1-, IFITM2-, or IFITM3-like based on their homology with the respective human IFITMs. Primate and rodent IFITMs cluster by isoform where IFITM1, IFITM2 and IFITM3 form separate groups, implying that these species arose after the three IFITM isoforms emerged by gene duplication. In contrast, IFITMs in all other species cluster in a species-specific manner, indicating that the separation of these species occurred before the ancestral IFITM diverged independently within each species. Notably, IFITMs in bats of the suborder Yangochiroptera and Yinpterochiroptera form two distinct monophyletic groups

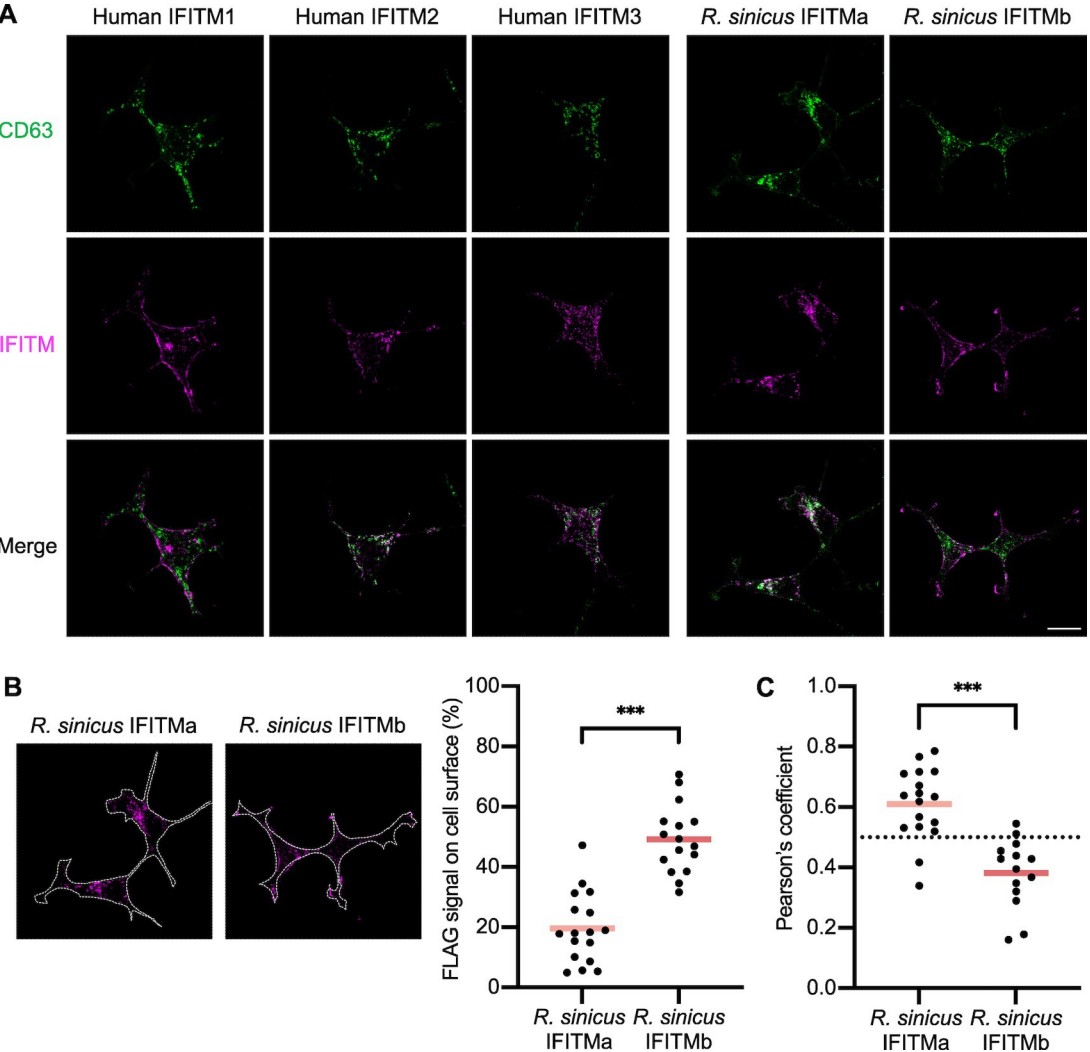

**Fig 4. Distinct subcellular localization of R. sinicus IFITMs. A.** HEK293T cells were transfected with the indicated FLAG-tagged IFITMs. Cells were stained for CD63 (green; late endosome marker) and FLAG (magenta; IFITMs) at 48 hours post-transfection and imaged by confocal microscopy. Representative z-stack images are shown. Scale bar, 30 μm. **B.** FLAG signal on the surface of each cell overexpressing the indicated FLAG-tagged IFITM (dotted line) was quantified by Fiji and expressed as a percentage of the total FLAG signal from the cell. Lines represent the mean from at least 30 cells. **C.** Pearson correlation coefficient analysis for FLAG-CD63 colocalization calculated with the JACoP plugin [96]. Lines represent the mean from at least 30 cells. Unpaired t-test; ***p<0.001.

with long branches, denoting that bat IFITMs have accumulated many mutations compared to the most recent common ancestor shared by all bat species.

Alternative splicing generates IFITM diversification as we have seen in *R. sinicus*. To understand how widespread alternative splicing is in the IFITM family, we identified *IFITM*-like genes in additional bat species from the RefSeq RNA database [76]. Analysis was restricted to 22 bat species due to limited datasets published at the time of analysis. Species were grouped by the pattern of alternative splicing in *IFITM*-like genes they possess (**Fig 7A–7B**). Alternatively spliced *IFITM*-like genes were evident in 12 bat species, but only 8 possess genes that encode non-synonymous IFITM isoforms, indicative of enhanced coding capacity. These bat species belong to a polyphyletic group, suggesting that they independently acquired

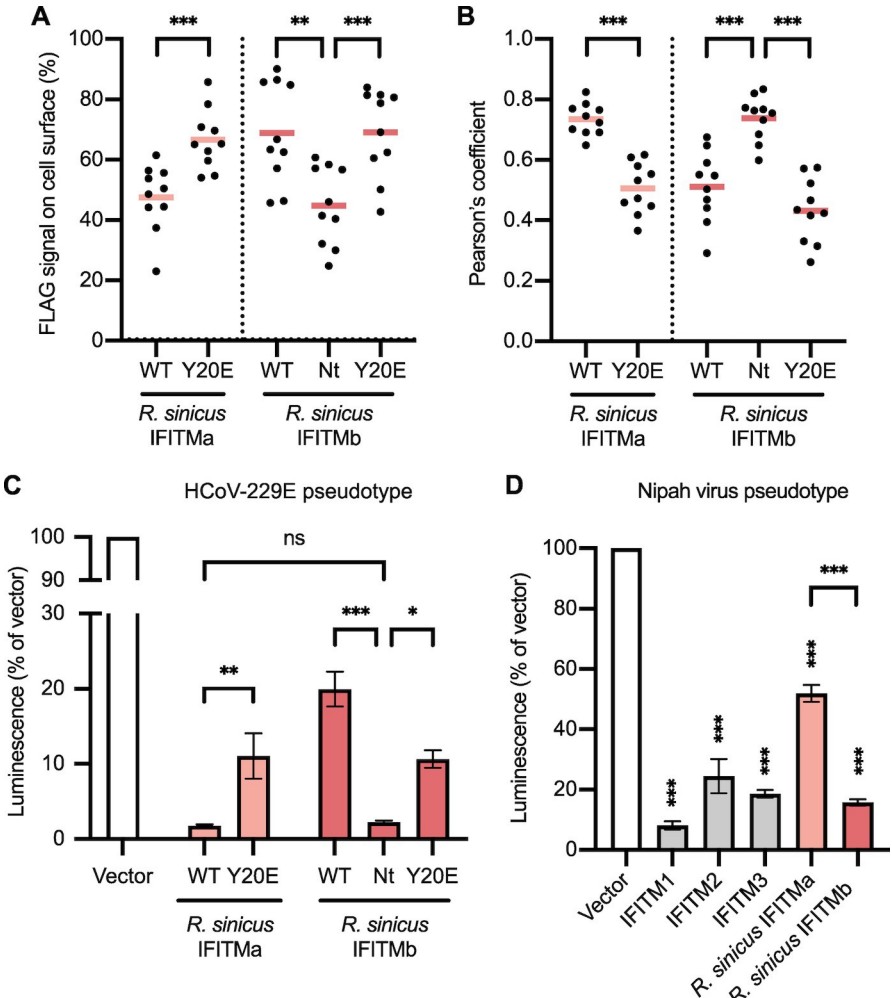

**Fig 5. Subcellular localization influences the antiviral activity of *R. sinicus* IFITMs. A.** HEK293T cells were transfected with the indicated FLAG-tagged IFITMs. Cells were stained for CD63 (green; late endosome marker) and FLAG (magenta; IFITMs) at 48 hours post-transfection and imaged by confocal microscopy. FLAG signal on the surface of cells was quantified by Fiji and expressed as a percentage of the total FLAG signal from the cell. Lines represent the mean from 10 images, each capturing 1–3 cells. **B.** Pearson correlation coefficient analysis for FLAG-CD63 colocalization calculated with the JACoP plugin [96]. Lines represent the mean from 10 images, each capturing 1–3 cells. **C.** HEK293T cells were co-transfected with the indicated IFITMs and APN then transduced with HCoV-229E pseudotypes encoding a luciferase reporter. Cells were lysed and analyzed by luciferase assay after 48 hours. Error bars represent SEM of averages from 3 independent experiments, each performed in triplicate. **D.** HEK293T cells were transfected with the indicated IFITMs and transduced with Nipah virus pseudotypes encoding a luciferase reporter. Cells were lysed and analyzed by luciferase assay after 48 hours. Error bars represent SEM of averages from 3 independent experiments, each performed in triplicate. Statistical significance of difference between vector- and IFITM-expressing cells were determined by one-way ANOVA with Dunnett's test; Statistical significance of difference between indicated groups were determined by unpaired t-test; *p<0.05, **p<0.01, ***p<0.001; ns, non-significant.

alternatively spliced *IFITM*-like genes. For 4 of these 8 species, alternative splicing occurs by alternative first exon and results in IFITM isoforms with distinct YXXΦ endocytic motif, suggestive of differential subcellular localization of these isoforms. This subset contained *R. sinicus* alongside three other bats, including the intermediate horseshoe bat *Rhinolophus affinis*. A new reference-quality genome of *R. affinis* was recently generated through the Bat1K project, from which we identified genes showing the highest homology with human *IFITM1-3* genes

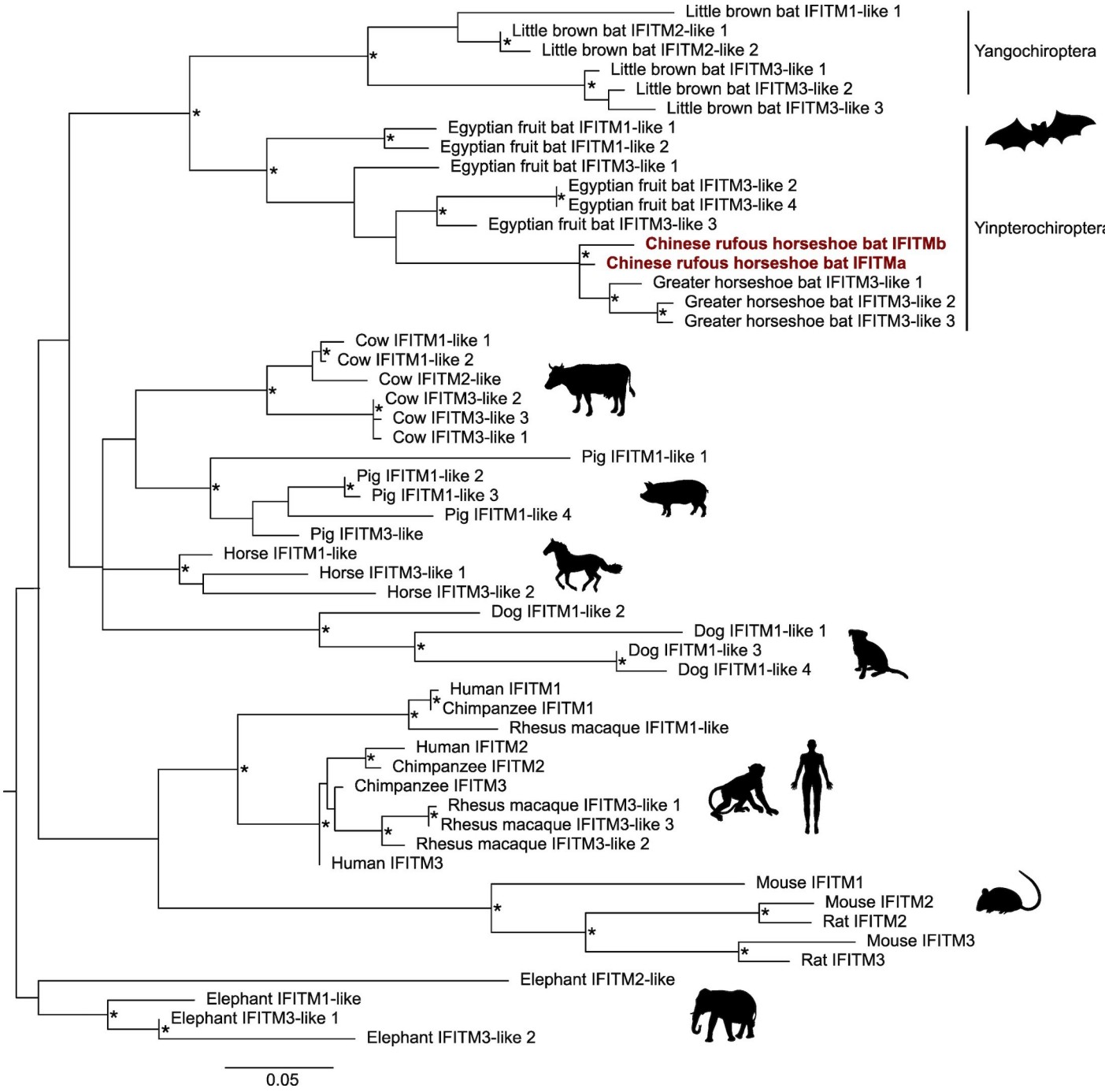

**Fig 6. Phylogenetic analysis of mammalian *IFITM*-like genes.** Phylogenetic tree constructed by maximum likelihood analysis of 54 IFITM protein-coding nucleotide sequences from different mammalian species. The tree was rooted on the elephant outgroup and nodes with bootstrap values >70% are marked with asterisks (*). Scale bar corresponds to 0.05 substitutions per site.

[77]. *R. affinis* possesses two *IFITM1*-like genes, one *IFITM2*-like gene and one *IFITM3*-like gene. The *R. affinis IFITM3* gene is predicted to encode two IFITM isoforms: IFITM3a with a $_{20}$DEML$_{23}$-containing N-terminus, and IFITM3b with a truncated N-terminus lacking the first 21 amino acid of IFITM3a (**S10 Fig**). As seen with *R. sinicus* IFITMa and IFITMb,

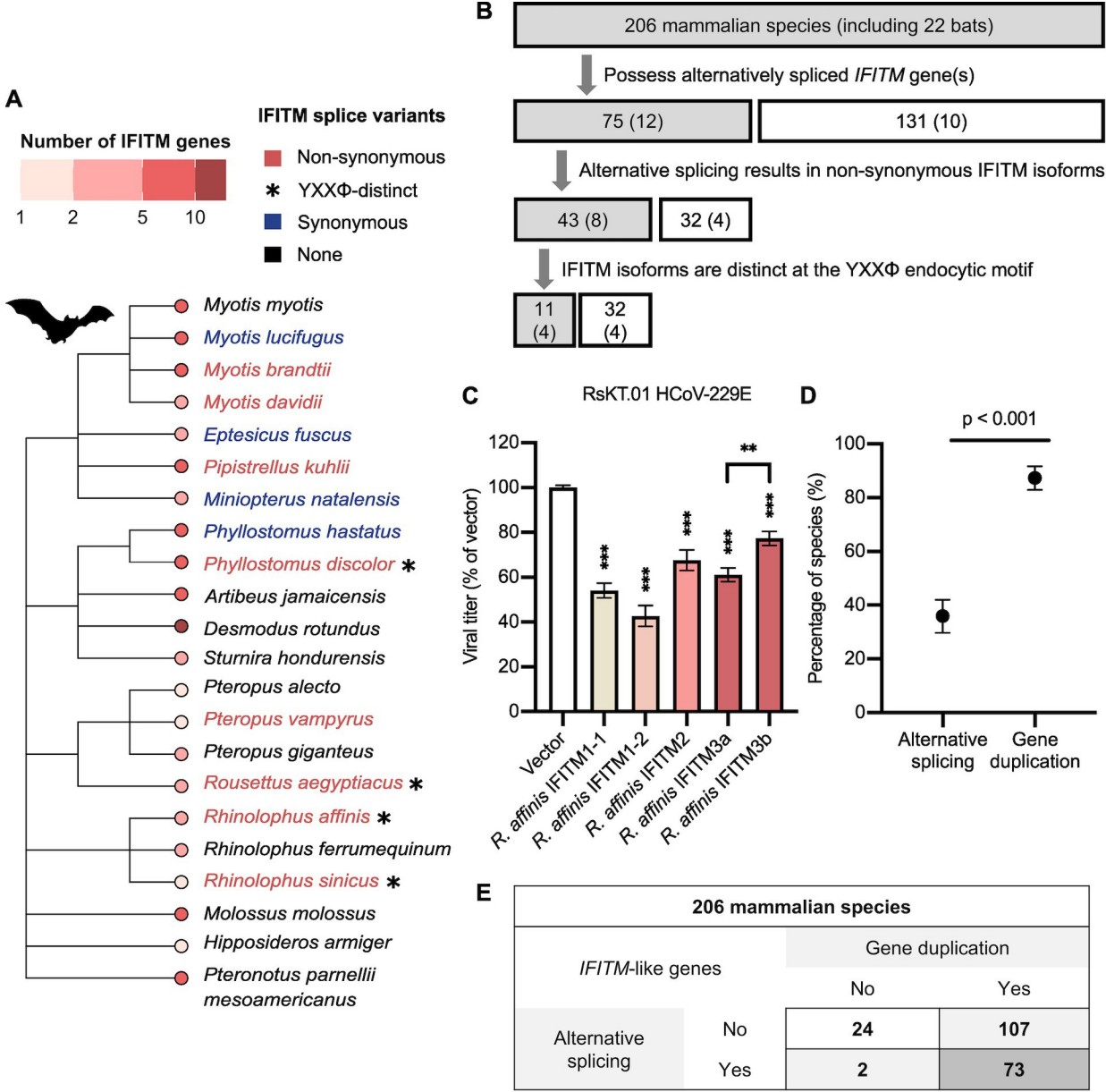

**Fig 7. Alternative splicing generates IFITM diversity in mammals. A.** Bats were grouped according to the *IFITM*-like genes they possess. A phylogenetic tree showing the ancestral relationships between bats was labelled by their grouping: bats with *IFITM*-like gene(s) that encode two or more synonymous (blue) or non-synonymous (red) IFITMs are colored. Bats with *IFITM*-like gene(s) encoding YXXΦ-distinct IFITM isoforms are marked with an asterisk (*). Tip nodes are colored by the number of *IFITM*-like genes they possess. **B.** Flow chart illustrating the classification of *IFITM* genes by their pattern of alternative splicing in 206 mammalian species, including 22 bats. **C.** RsKT.01 cells stably expressing the indicated *R. affinis* IFITMs were infected with HCoV-229E at MOI = 0.5. Infectious virus titers in the supernatants were determined by TCID50 assay. Error bars represent SEM of averages from 3 independent experiments, each performed in duplicate. Statistical significance of difference between vector- and IFITM-expressing cells were determined by one-way ANOVA with Dunnett's test; statistical significance of difference between indicated groups were determined by unpaired t-test; **p<0.01, ***p<0.001. **D.** Analysis of IFITM repertoires in 206 mammalian species revealed the frequency of alternative splicing and gene duplication of *IFITM*-like genes. Error bars represent the 95% confidence interval and statistical significance of difference was determined by bootstrapping with 1000 bootstraps. **E.** Association between alternative splicing and gene duplication in the *IFITM* family was tested by the Fisher's exact test.

*R. affinis* IFITM3a and IFITM3b may have distinct antiviral specificities. To confirm this, RsKT.01 cells stably expressing *R. affinis* IFITMs were challenged with HCoV-229E. All *R. affinis* IFITMs inhibited HCoV-229E infection with IFITM1-2 being the most potent inhibitor (**Fig 7C**). *R. affinis* IFITM3a and IFITM3b indeed exhibited differential antiviral potency, where IFITM3a inhibited HCoV-229E to a significantly greater extent. *R. affinis* therefore represents another example where IFITM functional diversity is generated by alternative splicing.

Finally, we extended the bioinformatic analysis to include all mammalian species in the RefSeq RNA database, leading to the identification of *IFITM*-like genes in an additional 184 species (**S11 Fig**). Only about 5% (11/206) of these mammals have *IFITM*-like genes that encode YXXΦ-distinct IFITM isoforms, including dog, ferret, meerkat, warthog, gelada baboon, brown bear and Angolan colobus monkey. Intriguingly, 13% (26/206) of sampled mammals have only one predicted IFITM transcript. We then compared alternative splicing with gene duplication as a strategy for IFITM diversification. Gene duplication is defined to have occurred in species that possess more than one *IFITM*-like gene, and it is not mutually exclusive with alternative splicing. *IFITM* gene duplication was observed in the majority of species and is more commonly adopted by mammals than alternative splicing (87% vs 36%) (**Fig 7D**). Notably, there is a strong association between the two strategies, meaning that species with multiple *IFITM*-like genes are more likely to also have *IFITM*-like genes that encode multiple transcripts (**Fig 7E**). Altogether, our evolutionary analysis uncovered alternative splicing as a previously underappreciated means of generating IFITM diversity in mammals.

## Discussion

Bats are natural reservoir hosts of pathogenic viruses with zoonotic potential, many of which are restricted by human IFITMs. Antiviral activity of human IFITM1-3 against these viruses has been widely studied, especially since the COVID-19 pandemic, but our understanding of IFITMs in species beyond human and mouse is limited. Our study demonstrates that *R. sinicus*, the Chinese rufous horseshoe bat, adopts alternative splicing to diversify its antiviral IFITM repertoire by generating two YXXΦ-distinct IFITM isoforms specialized to inhibit viruses that use different entry routes.

Using pseudotypes and replication-competent viruses, we show that while both *R. sinicus* IFITM isoforms have conserved amphipathic helices, they have distinct effects on virus infections. Distinct antiviral specificity is evident in the human IFITM family, with IFITM3 being the strongest inhibitor of IAV, whereas Marburg and Ebola filoviruses are more profoundly inhibited by IFITM1 [64]. This is thought to be explained by the different localization of IFITM1 and IFITM3, and the preferred entry routes of these viruses. We therefore predicted that a similar dynamic could underlie the differential antiviral potency observed between rsIFITMa and rsIFITMb against the studied viruses. We confirmed by fluorescence microscopy that rsIFITMa and rsIFITMb indeed localize to distinct cellular compartments. rsIFITMa with a YXXΦ-containing N-terminus co-localizes with the late endosomal marker CD63, consistent with the localization of microbat IFITM3 [46]. On the other hand, the surface localization of rsIFITMb closely resembles that observed for an N-terminally truncated murine IFITM3 which acts as a model for the human IFITM3 rs1225-C allele [78]. We further show, using N-terminal mutants, that cell surface-localized rsIFITMs are less able to inhibit HCoV-229E pseudotype entry compared to rsIFITMs that localized to endosomes. Nipah virus, which enters cells via direct fusion at the cell surface, is also more potently inhibited by rsIFITMb than rsIFITMa [72]. These findings suggest that the antiviral activity of rsIFITMs is influenced by their localization relative to the site of viral membrane fusion. Additional factors such as expression level, post-translational modifications and interactions with virus envelopes,

cellular co-factors, or other antiviral effectors may also influence the antiviral specificity and potency of IFITMs. Benfield *et al.* previously reported that mutation in codon 70 of microbat IFITM3 altered its localization and impaired its antiviral activity [47]. Intriguingly, codon 70 of *rsIFITMa* and *rsIFITMb* encodes a proline and a threonine respectively, possibly contributing to their functional differences. Overall, we propose that alternative splicing of IFITM is a means to generate distinctly localized IFITM isoforms that work together to block the two routes of viral entry, namely cell surface and endosomal entry.

Our phylogenetic analysis shows that in most cases, the ancestral *IFITM* gene was duplicated after speciation, so IFITMs in different species are not orthologs of one another. We further present here, with horseshoe bats as an example, that alternative splicing is a previously uncharacterized means of generating IFITM functional diversity in reservoir species. Alternative splicing in the IFITM family has not been widely reported apart from the human IFITM3 rs12252-C allele and the N-terminus truncated IFITM2 isoform found in human immune cells [52,53]. Nevertheless, our splicing analysis reveals that Chinese rufous horseshoe bats are not unique in using alternative splicing to generate IFITM diversity. This strategy is less commonly adopted than gene duplication in mammals but is still a significant contributor towards IFITM diversity. We also found that some mammals only express one IFITM transcript, which could imply that their *IFITM* family has not evolved by either mechanism. This may however result from limitations of our analysis as it is dependent on the quality of genome assemblies and RNA sequencing datasets. It is possible that limited IFITM diversity in some species is compensated by other antiviral effectors, or rich post-translational modifications that alter IFITM functions.

Alternative splicing is a source of phenotypic diversity in proteins beyond IFITMs, for example, the two splice variants of zinc finger antiviral protein (ZAPL and ZAPS) inhibit IAV infection via distinct mechanisms [79]. An unaddressed question is whether alternative splicing always leads to functional diversity [80]. Alternative first exon splicing regulates subcellular distribution of IFITMs and other proteins [81]. However, not all IFITM splice variants have distinct N-terminal domains as seen in Chinese rufous horseshoe bats and intermediate horseshoe bats. Functional characterization of IFITM splice variants in other mammals is necessary to understand the contribution of alternative splicing to the functional diversification of the IFITM family. It is also of interest to identify factors that influence expression levels of different gene transcripts at both the transcriptional and post-transcriptional levels. For instance, herpes simplex virus-1 evades immune restriction by favoring the expression of a splice variant of the antiviral myxovirus resistance protein 1 (MxA) that lacks antiviral activity [82]. Reservoir species may take advantage of alternative splicing to expand their antiviral IFITM repertoire, enabling them to carry a wider range of viruses asymptomatically. In the case of Chinese rufous horseshoe bats, we speculate that fine-tuning of the relative expression of rsIFITMa and rsIFITMb could provide specialized immunity against viruses.

Gene duplication and alternative splicing within the *IFITM* family are two strategies adopted by mammals for IFITM diversification–an example of convergent evolution. We functionally demonstrate here that IFITM alternative splicing in Chinese rufous horseshoe bats generates IFITM isoforms that are specialized in restricting viruses using different entry routes, thus conferring a broader antiviral coverage. For most mammals, the use of one or more strategies to expand the IFITM toolkit highlights the importance of IFITM diversity as a component of innate immunity. Our findings underscore the importance of transcriptomics when characterizing ISGs and suggest that studies that solely rely on a genomic approach could significantly underestimate the functional diversity of ISG families. Extending our splicing analysis and functional studies to other ISGs will reveal distinct antiviral transcriptomes in

different species–the result of evolution shaped by the virus-host arms race and the basis of species-specific zoonotic barriers.

## Materials and methods

### Identification of mammalian *IFITM* genes

*IFITM*-like genes in non-human species were identified by tBLASTn searches against the National Center for Biotechnology Information (NCBI) RefSeq RNA database using human IFITM3 as query [83]. The search was restricted to mammals (Taxonomy ID: 40674) and e-value cut-off was set to $1\times10^{-20}$ to exclude *IFITM5* and *IFITM10* orthologs. Non-coding RNAs were also excluded. The *R. sinicus* genome with NCBI RefSeq accession GCF_001888835.1 was used. Percentage identity between *R. sinicus* IFITMs were calculated using the pairwise sequence alignment tool EMBOSS Needle [84].

### RNA sequencing and analysis

RsKT.01 cells were seeded onto 6-well plates and either untreated, treated with 100 ng/ml HMW poly(I:C) (Invitrogen) for 6 hours or infected with VSV produced in Vero E6 cells at MOI = 0.1 overnight [63]. Cell lysates were collected for RNA extraction using the HP Total RNA kit (Omega Biotek). Ribosomal RNA was removed with Ribo-Zero rRNA depletion kit (Illumina) prior to cDNA synthesis using mixed oligoDT and random hexamer primers. Illumina next-generation sequencing was performed on the Novaseq 6000 system (2 x 250bp) by Novogene following their standard transcriptome protocol (including lncRNAs). Raw data were trimmed and assessed for quality using FastQC v0.11.8 [85]. Trimmed and cleaned reads were then mapped to the *R. sinicus* genome (Taxonomy ID: 89399) with STAR v2.7.2b [86]. Transcript abundances were quantified and normalized trimmed mean of M values (TMM) was calculated within the EdgeR v3.19 software package [87]. Differential expression analysis was performed within the same package. Cut-offs for differentially expressed transcripts were set to fold-change > 2 and p-value < 0.05. Relative expression of gene variants were calculated from isoform-specific reads.

### Structural characterization of peptides

Helical wheel projection plots of IFITM amphipathic helices were generated using the HELIQUEST software [88]. Amphipathic helix peptides were synthesized by Vivitide and reconstituted in DMSO to produce 4 mM stocks. For circular dichroism analysis, peptides were lyophilized and resuspended in 10 mM sodium borate (pH 7.4), 50 mM NaCl, 25 mM SDS, 3.3% ethanol and 50 mM NaCl at a final concentration of 170 μM. Spectra were acquired between 190 and 260 nm with continuous scanning at a rate of 20 nm/min on a Jasco J-1500 CD Spectropolarimeter. Spectra were recorded at 0.5 nm data pitch, 1 nm bandwidth and a digital integration time of 4 seconds. Secondary structure compositions of the peptides were determined using the BeStSel webserver [89].

### NBD-cholesterol binding assay

Binding of amphipathic helix peptides to NBD-cholesterol was assessed as described previously [16]. In brief, 500 nm NBD-cholesterol (Thermo Fisher, N1148) was incubated with serial dilutions of peptides (0–100 μM) in black-wall clear-bottomed 96-well plates. Plates were incubated for 1 hour at room temperature or 4°C before measuring fluorescence intensity and polarization respectively. Measurements were taken by a Tecan Infinite M1000 at 470 nm excitation and 540 nm emission. A rotavirus NSP4-derived peptide was used as negative control.

## Cell culture

HEK293T (ATCC and ECACC), HEK293T-ACE2-TMPRSS2 (NIBSC), Huh7 and Huh7.5 (gift from Prof. Jürgen Haas, University of Edinburgh), and RsKT.01 (immortalized *R. sinicus* kidney epithelial cells developed in our lab [63]) cells were cultured in DMEM with GlutaMAX (Thermo Fisher), supplemented with 10% fetal bovine serum and 1% penicillin-streptomycin (Gibco) at 37˚C and 5% $CO_2$ [63]. Media was supplemented with non-essential amino acids for Huh7.5 cells and normocin (Invitrogen) for RsKT.01 cells.

## Transfection and western blotting

rsIFITMa and rsIFITMb constructs with a FLAG tag at the N-terminus were synthesized by IDT or Beijing Tsingke Biotechnology and cloned into the pQCXIP vector for transfection. HEK293T and Huh7.5 cells were transfected with Lipofectamine 2000 (Invitrogen) and FuGENE HD (Promega) reagent respectively, at a 1:3 ratio of plasmid DNA to transfection reagent according to manufacturer's protocol. For IFITM upregulation in RsKT.01 cells, cells were transfected with 1 μg/ml HMW poly(I:C) (Invivogen) in 6-well plates using FuGENE 6 (Promega) as per the manufacturer's protocol. At 48 hours post-transfection, cells were lysed in a 1% Triton X-100 buffer supplemented with 1x NuPAGE LDS sample buffer (Invitrogen), 50 U/ml benzonase (Merck), and phosphatase and protease inhibitors (Roche or Phygene). Lysates were boiled with 50 mM DTT at 95˚C for 5 minutes. Protein was separated by 4–12% Bis-Tris protein gel (Bio-Rad or Yeasen) and transferred to 0.2 μm PVDF membrane (Cytiva). The following antibodies and dilutions were used: mouse anti-FLAG-M2 (Sigma; 1:20,000), mouse anti-IFITM1 (Proteintech; 1:50,000), rabbit anti-IFITM2 (Cell Signalling Technology; 1:1,000), rabbit anti-fragilis (Abcam; 1:5,000), mouse anti-beta-actin (Santa Cruz Biotechnology; 1:1,000), rabbit anti-alpha-tubulin (Proteintech; 1:10,000), mouse anti-GAPDH (Proteintech; 1:1,000), IRDye 800CW goat anti-mouse (LICOR; 10,000), IRDye 800CW goat anti-rabbit (LICOR; 1:10,000), HRP-conjugated goat anti-rabbit IgG (Abcam; 10,000), HRP-conjugated goat anti-mouse IgG (Abcam; 10,000). Protein was detected on a Li-Cor Odyssey imaging system or C-DiGit blot scanner.

## Stable cell line production

To generate RsKT.01 cells stably expressing *R. sinicus* or *R. affinis* IFITMs, IFITM constructs were synthesized by Tsingke Biotech and cloned into the pLVX-IRES-mCherry vector for lentivirus production by co-transfecting HEK293T cells with the psPAX2 plasmid. Lentiviral supernatant was passed through a 0.45 μm filter and added to RsKT.01 cells with 4 μg/ml polybrene (Biosharp) for 4–6 hours. At 72 hours post-transduction, cells were sorted for mCherry fluorescence and pooled to culture as stable cell lines. To generate HEK293T cells stably expressing human or *R. sinicus* IFITMs, pQCXIP-IFITM constructs were used for retrovirus production with the CMVi packaging plasmid kindly gifted by Prof. Greg Towers (University College London) [90]. Retroviral supernatant was passed through a 0.45 μm filter and added to HEK293T cells with 4 μg/ml polybrene (Merck). At 72 hours post-transduction, cells were selected in 2 μg/ml puromycin and cultured as stable cell lines after protein expression was validated by western blotting.

## CRISPR/Cas9-mediated knockout

*rsIFITMa* knockout cells were generated using a lentivirus-based system for Cas9/guide RNA expression. Lentiviruses were generated by transfecting producer HEK293T cells with psPAX2, lentiCRISPRv2, pRSV-REV, pMDL-VSV-G and pCRISPRia-v2 encoding a blue

fluorescence protein and guide RNA. The guide RNA used has the sequence CTGCGGATGT-TAACCACGG and targets exon 1 of *rsIFITMa*. Lentiviral supernatant was passed through a 0.45 μm filter and added to RsKT.01 cells with 4 μg/ml polybrene (Biosharp) for 4–6 hours. Media was replaced after 6 hours. At 48 hours post-transduction, cells were selected in 2.5 μg/ml puromycin for one week before further selecting for BFP-positive cells by fluorescence activated cell sorting (FACS) and validating protein expression by western blotting.

## Replication-competent virus infection

HEK293T cells were seeded onto 24-well plates one day prior to infection with Influenza virus A/Puerto Rico/8/1934 (H1N1, PR8) that was either purchased from Charles River Laboratories or kindly gifted by Prof. Paul Digard (University of Edinburgh). Cells were infected at the indicated MOIs for 18 hours at 37˚C for single-cycle infections, or at MOI = 0.05 for multi-cycle infections. Huh7.5 cells were infected with HCoV-229E-GFP kindly gifted by Prof. Volker Thiel (University of Bern) at MOI = 0.05 for 18 hours at 34˚C. To measure infection, cells were fixed and permeabilized with the Cytofix/Cytoperm kit (BD Biosciences), immunostained using 1:500 rabbit anti-FLAG (Sigma) and mouse anti-IAV NP (Abcam), followed by 1:300 Alexa Fluor 488-conjugated goat anti-mouse (Invitrogen) and Alexa Fluor 647-conjugated goat anti-rabbit (Invitrogen), and analyzed on a LSRFortessa flow cytometer (BD Biosciences). FLAG immunostaining was not included for infections in stably transduced cell lines. Flow cytometry data were analyzed using the online tool Floreada.io (https://floreada.io/analysis) and FlowJo software (BD Life Sciences).

RsKT.01 cells stably expressing IFITMs were infected with HCoV-229E produced in Huh7 cells at MOI = 0.5 in media supplemented with 1% FBS at 37˚C. Infectious virus titers in the supernatants were quantified using 50% tissue culture infectious dose ($TCID_{50}$) assay performed in triplicates. Briefly, $3 \times 10^4$ Huh7 cells were seeded in each well of 96-well plates and incubated overnight to obtain a confluent cell layer. The following day, cells were inoculated with 1:10 serially diluted viral supernatants and incubated at 37˚C. After 2 hours, viral supernatants were removed and 100 μl of complete media supplemented with 10% fetal bovine serum was added to the wells. The plates were incubated at 37˚C for 3–5 days before observing cytopathic effects under a light microscope. $TCID_{50}$/ml was calculated using the Spearman and Karber algorithm. Where indicated, infection of vector-expressing cells was set to be 100% to allow normalization and comparison across experiments.

## Lentiviral-based pseudotyped viruses production and transduction

SARS-CoV and MERS-CoV pseudotypes were produced by co-transfecting HEK293T cells with pNL4-3.LucR⁻E and pcDNA3.1 plasmids containing SARS-CoV or MERS-CoV spike. HCoV-229E, SARS-CoV-2 and Nipah pseudotypes were produced by co-transfecting HEK293T cells with p8.91 Gag-Pol, pCSFLW and pcDNA3.1 plasmids encoding HCoV-229E or SARS-CoV-2 Wuhan/Delta/Omicron spike, or two plasmids encoding Nipah F and G glycoproteins respectively [91,92]. Plasmids encoding Nipah virus glycoproteins were kindly gifted by Dr. Edward Wright (University of Sussex). Supernatant was harvested at 72 hours post-transfection, passed through a 0.45 μm filter and stored at -80˚C. Target cells that transiently express the respective virus entry receptor were incubated with pseudotypes for 48 hours, pseudotype entry was then quantified using Bright-Glo luciferase assay (Promega). For experiments involving drug pre-treatment, target cells were incubated with DMSO or MDL-28170 (20 μM) for 1 hour prior to transduction. Luminescence of transduced vector-transfected cells was set to be 100% to allow normalization and comparison across experiments.

## Negative imprinting assay

HEK293T cells stably expressing the indicated IFITMs were used as producer cells to produce IAV pseudotypes. Cells were co-transfected with p8.91 Gag-Pol, pCSFLW, pPolII-H5HA and pI.18-HAT. At 24 hours post-transfection, 0.25 U/ml exogenous neuraminidase (Sigma) was added to the media. Supernatant was harvested at 72 hours post-transfection, passed through a 0.45 μm filter and viral titer was determined by product-enhanced reverse transcriptase (PERT) assay as described previously [93,94]. The PERT assay determines the amount of exogenous RNA template that is converted into complementary DNA by lentiviral reverse transcriptase (RT) in the viral supernatant. In brief, viral supernatants were lysed and used as input for the PERT assay, along with serially diluted recombinant RT to generate a standard curve for absolute quantification. RT-normalized viral supernatants were then used to transduce HEK293T cells and pseudotype entry was quantified using Bright-Glo luciferase assay (Promega) after 48 hours.

## Absolute quantification of IFITM mRNA abundance

HEK293T cells were transfected with rsIFITMa or rsIFITMb at the indicated dosages in 24-well plates. RsKT.01 cells were transfected with 1 μg poly(I:C) in 6-well plates to induce IFITM expression. After 48 hours, cells were harvested and RNA was extracted using the E.Z. N.A. HP Total RNA Kit (Omega Bio-tek). RT-qPCR was performed using the following primers: rsIFITMa forward, ACCGTGGTTAACATCCGCAG; rsIFITMa reverse, CCGGTCCCTA GACTTCACGG; rsIFITMb forward, CACCCAGACTCTCACTCTCAG; rsIFITMb reverse, CGGTGCATCTCTGGCGTTC. Copy numbers of IFITM3 per 1 ng of RNA were determined by normalizing Cq values against a standard curve generated by titration of IFITM gene DNA cut and excised from the plasmids, normalized by molecular weight.

## Confocal immunofluorescence microscopy

Cells transiently expressing IFITMs were seeded onto 8-well chamber slides (ibidi) at 50,000 cells per chamber. After 24 hours, cells were fixed and permeabilized with the Cytofix/Cytoperm kit (BD Biosciences), immunostained using 1:400 mouse anti-CD63 (Santa Cruz Biotechnology) and rabbit anti-FLAG (Sigma), followed by 1:300 Fluor 488-conjugated goat anti-mouse (Invitrogen) and Alexa Fluor 647-conjugated goat anti-rabbit (Invitrogen). Slides were mounted with ProLong glass antifade mountant (Invitrogen) and imaged on the Leica TCS SP8 confocal microscope. Z-stack processing and further analyses were performed in Fiji [95]. Pearson's coefficient for FLAG-CD63 colocalization was calculated with the JACoP plugin [96].

## Construction of IFITM phylogenetic tree

Phylogenetic analysis of mammalian IFITMs was performed in MEGA X [97]. To exclude non-functional IFITMs, the following criteria were used for IFITM inclusion: protein length of 102–157 amino acids; contains 2 exons; presence of CD225 and a transmembrane domain [61,98–100]. The best-fit nucleotide substitution model was determined by maximum likelihood analysis and a tree was constructed from IFITM protein-coding nucleotide sequences with 1000 bootstraps. The tree was annotated in FigTree [101].

## *IFITM* alternative splicing analysis

Alternative splicing patterns of mammalian *IFITM*-like genes were characterized by analyzing mRNA transcripts produced from identified mammalian *IFITM*-like genes using NCBI Datasets [102]. Transcripts encoded by the same gene were compared by nucleotide alignment and

genes were grouped based on their splicing pattern. Phylogenetic tree inferring the evolutionary relationships between analyzed species was generated using the NCBI taxonomy browser [103]. Code scripts for the analysis are deposited at https://github.com/nellymak1/Mammalian_IFITM_splicing_analysis.

## Supporting information

**S1 Fig. Alignment and identification of interferon-stimulated response elements (ISRE) in human and *R. sinicus* IFITMs. A.** Identification of interferon-stimulated response elements (ISRE) around the transcription start site (± 350 base pairs) of human and *R. sinicus IFITM* genes. ISREs were defined to have the consensus sequence of GAAANNGAAA or TTTCNNTTTC, with no mismatch (red) or 1 mismatch (blue) [104]. **B.** Amino acid sequence alignment of human IFITM1-3 and *R. sinicus* IFITMs. Protein domains, functional motifs and amino acids that undergo post-translational modifications are highlighted. Asterisks (\*) indicate positions with a conserved residue; colons (:) and periods (.) indicate conservation between groups of strongly and weakly similar properties respectively. Percentage identity was calculated using the pairwise sequence alignment tool EMBOSS Needle [84].
(TIF)

**S2 Fig. Expression of rsIFITM isoforms in RsKT.01 cells. A.** Absolute quantification of IFITM mRNA abundance was performed by RT-qPCR. Untreated and poly(I:C)-treated RsKT.01 cells, and HEK293T transfected with *R. sinicus* IFITMa or IFITMb constructs, were subjected to RNA extraction and RT-qPCR with isoform-specific primers. Exact copy numbers per ng of RNA were determined by normalizing Cq values against a standard curve. Error bars represent SEM of averages from 3 independent experiments, each performed in triplicate. **B.** RsKT.01 cells were untreated or transfected with poly(I:C), followed by detection of IFITM expression by western blotting at 24 and 48 hours post-transfection. Representative western blot from 3 independent experiments is shown. **C.** Quantitative analysis of the IFITM/GAPDH ratio from western blots. Error bars represent SEM of averages from 3 independent experiments. **D.** Knockout cells were generated by CRISPR/Cas9-mediated knockout using a guide RNA that targets the first exon of *rsIFITMa*. Wild-type cells and *rsIFITMa* knockout cells were transfected with poly(I:C), followed by detection of IFITM expression by western blotting at 48 hours post-transfection.
(TIF)

**S3 Fig. Characterization of *R. sinicus* IFITM amphipathic helix. A.** Helical wheel project plots of human and *R. sinicus* IFITM amphipathic helices containing hydrophobic (yellow) and hydrophilic (purple or pink) residues. Arrows indicate magnitude and direction of mean hydrophobic moments. **B.** Mean hydrophobic moment of IFITM amphipathic helices from 34 mammalian species, including 19 bat species. Medians of each group are shown. Kruskal-Wallis test; \*p<0.05, \*\*\*p<0.001. **C.** NBD-cholesterol fluorescence polarization was measured following incubation of IFITM amphipathic helix peptides (0–50 μM) with NBD-cholesterol (500 nM). Data points are normalized to 50 uM human IFITM2/3. Error bars represent SEM of averages from 3 independent experiments. Statistical significance of differences between human IFITM2/3 and another peptide was determined by one-way ANOVA with Dunnett's test; \*p<0.05, \*\*p<0.01. **D-F.** HEK293T cells were transfected with FLAG-tagged IFITM3 or chimeric IFITM3-AH [*R. sinicus*] as illustrated in (D), and protein expression was detected by western blotting at 24 hours post-transfection (E). Transfected cells were infected with IAV at MOI = 0.05 and analyzed by flow cytometry at 18 hours post-infection. Error bars represent SEM of averages from 3 independent experiments, each performed in duplicate. Statistical

significance of difference between vector- and IFITM-transfected cells was determined by one-way ANOVA with Dunnett's test; Statistical significance of difference between IFITM3- and IFITM3-AH [*R. sinicus*]-transfected cells was determined by unpaired t-test; ***p<0.001; ns, non-significant.
(TIF)

**S4 Fig. Flow cytometry analyses of virus restriction by transfected IFITMs. A-B.** Huh7.5 (A) or HEK293T (B) cells were transfected with FLAG-tagged IFITM constructs and infected with HCoV-229E-GFP or IAV for 18 hours respectively. Flow cytometry dot plots show the gating strategy to identify cells that are double positive for FLAG staining and GFP or IAV nucleoprotein (NP) staining. Percentages of gated cells within the parent population are shown. **C.** Percentage of successfully transfected cells were determined by gating for FLAG. Error bars represent SEM of averages from 3 independent experiments. EV, empty vector.
(TIF)

**S5 Fig. Distinct antiviral potency of *R. sinicus* IFITMs under various experimental conditions. A.** HEK293T cells were stably transduced to express the indicated IFITMs. IFITM expression was detected by western blotting. **B.** Stably transduced HEK293T cells were infected with IAV at the indicated MOIs and analyzed for NP staining by flow cytometry at 18 hours post-infection. Error bars represent SEM of averages from 3 independent experiments, each performed in duplicate. **C.** Stably transduced HEK293T cells were infected with IAV at MOI = 0.05 and analyzed for NP staining by flow cytometry at the indicated time points. Error bars represent SEM of averages from 3 independent experiments, each performed in duplicate. **D.** IAV pseudotypes encoding a luciferase reporter were produced from HEK293T cells stably expressing the indicated IFITMs. RT-normalized IAV pseudotypes were used to transduce wild-type HEK293T cells to determine infectivity. Cells were lysed and analyzed by luciferase assay after 48 hours. Error bars represent SEM of averages from 3 independent experiments. Statistical significance of difference between vector- and IFITM-expressing cells was determined by one-way ANOVA with Dunnett's test; statistical significance of difference between *R. sinicus* IFITMa- and IFITMb-expressing cells was determined by unpaired t-test; *p<0.05, **p<0.01.
(TIF)

**S6 Fig. Flow cytometry analyses of virus restriction by stably expressed IFITMs. A.** HEK293T cells were stably transduced to express the indicated FLAG-tagged IFITMs. Histograms show IFITM expression assessed by FLAG staining and the percentage of FLAG-positive cells. **B.** HEK293T cells stable expressing FLAG-tagged IFITMs were infected with IAV. Flow cytometry dot plots show the gating strategy to identify cells that are positive for IAV NP staining. Percentages of gated cells within the parent population are shown.
(TIF)

**S7 Fig. *R. sinicus* IFITMs exhibit dose-dependent antiviral potency. A.** HEK293T cells were transfected with the indicated amount of FLAG-tagged *R. sinicus* IFITMa or IFITMb in 24-well plates. The total amount of transfected DNA was kept constant with an empty plasmid. IFITM expression was determined by western blotting at 24 hours post-transfection. **B.** HEK293T cells were co-transfected with the indicated amount of rsIFITMa or rsIFITMb and APN then transduced with HCoV-229E pseudotypes encoding a luciferase reporter. Cells were lysed and analyzed by luciferase assay after 48 hours. Error bars represent SEM of averages from 3 independent experiments, each performed in triplicate. Statistical significance of difference between vector- and IFITM-expressing cells were determined by one-way ANOVA with

Dunnett's test; ***p<0.001.
(TIF)

**S8 Fig. Expanded panel of immunofluorescence images of *R. sinicus* IFITMs.** HEK293T cells were transfected with FLAG-tagged *R. sinicus* IFITMa or IFITMb. Cells were stained for CD63 (green; late endosome marker) and FLAG (magenta; IFITMs) at 48 hours post-transfection and imaged by confocal microscopy. Representative z-stack images are shown. Scale bar, 30μm.
(TIF)

**S9 Fig. The effect of virus entry route on IFITM sensitivity. A.** Schematic showing *R. sinicus* IFITMa and IFITMb N-terminal mutants. **B.** HEK293T cells were transfected with FLAG-tagged *R. sinicus* IFITMs with the indicated mutations. IFITM expression was detected by western blotting at 24 hours post-transfection. **C.** HEK293T cells were transfected with the indicated FLAG-tagged IFITMs. Cells were stained for CD63 (green; late endosome marker) and FLAG (magenta; IFITMs) at 48 hours post-transfection and imaged by confocal microscopy. Representative z-stack images are shown. Scale bar, 15 μm. **D.** HEK293T and HEK293-T-ACE2-TMPRSS2 cells were transfected with APN and transduced with HCoV-229E pseudotypes in the presence of DMSO or MDL-28170. Cells were lysed and analyzed by luciferase assay after 48 hours. Data points were normalized to the respective DMSO-treated cells (white bars). Error bars represent SEM of averages of 3 independent experiments, each performed in triplicate. **E.** HEK293T-ACE2-TMPRSS2 cells were co-transfected with APN and the indicated FLAG-tagged IFITM constructs, then transduced with HCoV-229E pseudotypes in the presence of DMSO or MDL-28170. Cells were lysed and analyzed by luciferase assay after 48 hours. Data points were normalized to vector/DMSO. Error bars represent SEM of averages from 3 independent experiments, each performed in triplicate. Statistical significance of difference between indicated groups were determined by unpaired t-test; *p<0.05, **p<0.01, ***p<0.001.
(TIF)

**S10 Fig. Sequence alignment of *R. affinis* IFITMs.** Protein sequence alignment of *R. affinis* IFITMs that show highest homology with human IFITM1-3. Location of the endocytic motif is highlighted. Asterisks (*) indicate positions with a conserved residue; colons (:) and periods (.) indicate conservation between groups of strongly and weakly similar properties respectively.
(TIF)

**S11 Fig. Extended splicing analysis of mammalian *IFITM*-like genes.** Analysis of *IFITM*-like genes in Fig 7A is extended to include 206 mammalian species. Mammals were grouped according to the *IFITM*-like genes they possess. Phylogenetic tree showing the ancestral relationships between these species was labeled by their grouping: species with *IFITM*-like gene(s) that encode two or more synonymous (blue) or non-synonymous (red) IFITMs are colored. Species with *IFITM*-like gene(s) encoding YXXΦ-distinct IFITM isoforms are marked with an asterisk (*). Tip nodes are colored by the number of *IFITM*-like genes they possess.
(TIF)

**S1 Data. Raw values for plots displayed in this manuscript.**
(XLSX)

**S2 Data. List of accession numbers of *IFITM*-like genes included in Fig 6.**
(XLSX)

**S3 Data. Data for generating plots in Figs 7A and S11.**
(XLSX)

## Acknowledgments

We would like to thank Dr. Marjolein Kikkert (Leiden University Medical Center) for useful discussions, Dr. Rob Young (University of Edinburgh) for guidance on the bioinformatic analyses, Maria Sole Regina Lancerin (University of Edinburgh) for help with the PERT assay, and Dr Connor Bamford (Queen's University Belfast) for helpful comments on the manuscript. We also thank Prof. Jürgen Haas (University of Edinburgh), Prof. Volker Thiel (University of Bern) and Prof. Jincun Zhao (Guangzhou Institute of Respiratory Health) for cells and HCoV-229E viruses.

## Author Contributions

**Conceptualization:** Nelly S. C. Mak, Aaron T. Irving, Alex A. Compton, Richard D. Sloan.

**Data curation:** Nelly S. C. Mak.

**Formal analysis:** Nelly S. C. Mak, Jingyan Liu, Dan Zhang, Xiaomeng Li, Feiyu Chen, Aaron T. Irving.

**Funding acquisition:** Aaron T. Irving, Alex A. Compton, Richard D. Sloan.

**Investigation:** Nelly S. C. Mak, Jingyan Liu, Dan Zhang, Kazi Rahman, Feiyu Chen, Siddhartha A. K. Datta, Kin Kui Lai, Aaron T. Irving.

**Methodology:** Nelly S. C. Mak, Jingyan Liu, Dan Zhang, Kazi Rahman, Siddhartha A. K. Datta, Nigel Temperton.

**Resources:** Zhengli Shi, Nigel Temperton, Aaron T. Irving, Alex A. Compton, Richard D. Sloan.

**Software:** Nelly S. C. Mak, Jordan Taylor, Xiaomeng Li.

**Supervision:** Alex A. Compton, Richard D. Sloan.

**Validation:** Nelly S. C. Mak.

**Visualization:** Nelly S. C. Mak, Jordan Taylor.

**Writing – original draft:** Nelly S. C. Mak.

**Writing – review & editing:** Jordan Taylor, Nigel Temperton, Aaron T. Irving, Alex A. Compton, Richard D. Sloan.

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
