## [Decision Letter · Decision Letter 0]

5 Mar 2024

Dear Sloan,

Thank you very much for submitting your manuscript "Alternative splicing expands the antiviral IFITM repertoire in Chinese horseshoe bats" for consideration at PLOS Pathogens. As with all papers reviewed by the journal, your manuscript was reviewed by members of the editorial board and by several independent reviewers. In light of the reviews (below this email), we would like to invite the resubmission of a significantly-revised version that takes into account the reviewers' comments.

While three reviewers noted your study focus and rational, they split in respect to strength of evidence and scale of advancement. Addressing these concerns, as detailed by two reviewers in the Major Issues below, will be critical for our ultimate decision. 

Additionally and in line with the respective reviewers' comments, please improve accuracy of the taxonomic description of bats and viruses of the study across the manuscript. For instance, please consider  "many Chinese horseshoe bat species" instead  of "Chinese horseshoe bats" or alike in the Title and elsewhere. Likewise, consider "species *Severe acute respiratory syndrome-related coronavirus*" instead of "SARS coronaviruses" in references to the respective viruses infecting one or more bat species in the field. Consult https://doi.org/10.1038/s41564-020-0695-z for guidance on the nomenclature of coronavirus taxa that include SARS-CoV and SARS-CoV-2.      

We cannot make any decision about publication until we have seen the revised manuscript and your response to the reviewers' comments. Your revised manuscript is also likely to be sent to reviewers for further evaluation.

Sincerely,

Alexander Gorbalenya

Section Editor

PLOS Pathogens

Michael Malim

Editor-in-Chief

PLOS Pathogens

orcid.org/0000-0002-7699-2064

Reviewer's Responses to Questions

**Part I - Summary**

Reviewer #1: In this manuscript by Mak et. al., the authors seek to characterize the expression and antiviral activity of IFITMs in Rhinolophus bat species (R. sinicus), which are important reservoirs for zoonotic viruses. They provide compelling data that R. sinicus bats encode different IFITM isoforms through alternative splicing. Further, they show that these isoforms have antiviral effects when exogenously overexpressed. The authors expand their survey to transcriptomes of diverse mammalian species and find that alternative splicing is an important feature of IFITM diversification. The manuscript is clearly written, and experiments are well controlled. While this work will be of interest to the field, I do feel that the authors conclusions could be strengthened with additional experiments and analyses. My major and minor comments are listed as follows.

Reviewer #2: In the manuscript written by Mak et al. data are presented suggesting alternative splicing of IFITM genes in Chinese horseshoe bats (Rhinolophus sinicus), resulting in two constitutively expressed IFITM isoforms, RsIFITMa and RsIFITMb, as shown at the RNA level. Proteins expressed from cloned cDNAs from both RNA isoforms differ in subcellular localisation and antiviral profile. Looking at 206 mammalian species, the authors claim that alternative splicing is a ubiquitous strategy for evolutionary diversification of the IFITM gene family. While the data are interesting, there is no evidence provided of native protein expression from the identified alternatively spliced transcripts, the depth of the antiviral assays is not very profound, the gain of knowledge is at best incremental and a more specialised journal might better suited for publication.

Reviewer #3: In the manuscript titled “Alternative splicing expands the antiviral IFITM repertoire in Chinese horseshoe bats,” Mak and colleagues explore the expression of two isoforms of an IFITM gene in the Chinese horseshoe bat through alternative splicing. They conducted experiments overexpressing these isoforms in both human and bat cell lines, demonstrating their antiviral activity, with one isoform showing superior activity than the other. Additionally, the team analyzed the IFITM genes in 206 mammals, revealing that alternative splicing serves as a mechanism for IFITM diversification in 75 species, in contrast to gene duplication, observed in 180 species. This study significantly contributes to our understanding of the IFITM-mediated innate immunity in bats, potentially elucidating the species' resilience to certain viruses. The experiments are meticulously conducted, and the manuscript is well written. Nevertheless, the manuscript could be enhanced by addressing the following points.

**Part II – Major Issues: Key Experiments Required for Acceptance**

Reviewer #1: 1) Fig 3: Different MOIs may lead to different outcomes. Testing the robustness of rsIFITM antiviral activity at different MOIs should be considered. The corresponding flow data (Fig. S3) shows that the authors claims are being made based on a small population of infected cells (~10% in control cells), so I question whether this phenotype is abolished at slightly higher MOIs. It would also be worth evaluating the kinetics of virus spread -/+ rsIFITMs (e.g., growth curve experiments with replication competent viruses) rather than single stagnant timepoints.

2) Fig 3: The authors should demonstrate how the magnitude of rsIFITM expression in transfected cells relates to endogenous expression in bat cell lines. For instance, in the rsIFITM transfected cells, is the mRNA abundance anywhere close to what was observed following ISG induction in bat cell lines? The authors should also consider transfecting cells with varying amounts of plasmid DNA and measuring dose effects on virus infectivity. Exceedingly high levels of overexpressed IFITMs may not accurately reflect the true natural state of the protein.

3) Fig 3G: It was a nice addition to perform experiments in a relevant bat cell line. However, this is still quite artificial as the authors are relying on exogenous overexpression of rsIFITMs, and their phenotypes for 229E are still quite modest. To make claims that they are “antiviral in their native cell background” the authors would need to either knockdown (or knockout) these isoforms in bat cells and then assess virus infection. Even more important would be to determine whether knockdown (or knockout) in the presence of ISG induction (e.g., interferon stimulation) has any effects on infection outcomes.

4) Fig. 4 The analyses of “FLAG signal on the cell surface” lack robustness. Line 888-889 “averages from 3 cells” is insufficient. The authors need to increase their cell number by imaging a set number of random fields of view. Using a membrane marker stain would strengthen this analysis. Quantification of CD63-FLAG colocalization would also be useful. As a caveat, it’s important to note that artificial overexpression may change the localization of IFITM proteins.

5) A western blot accompanying Fig. 6C would be important to show that differences in expression are not accounting for the different phenotypes observed. Also, given the modest phenotypes (<2 fold reduction at most), testing another virus isolate (or pseudovirus) would help add to the claim of “distinct antiviral specificities” between the different isoforms.

Reviewer #2: 1. Please provide more details on RNA sequencing. The authors acknowledge that ‘absolute quantification of expression was not possible due to biased handling of reads which mapped to sequences shared by both isoforms’. How were the reads unambiguously assigned to one or the other isoform then (Fig. 1B, C, D)?

2. There is no evidence provided that proteins are indeed being generated and stably expressed from both native transcripts. All structural data are based on predictions and on analysis of synthesised peptides corresponding to merely the amphipathic helices.

3. Figure 3: please provide raw flow cytometry data showing the gating strategy and the specificity of the anti-FLAG immunostaining

4. Figure 4: the rationale to test the impact of the RsIFITMa and RsIFITMb in RsKT.01 cells which already express endogenous levels of both (as shown in Fig. 1) is unclear – the antiviral activity of the heterologously expressed proteins might be underestimated due to the contribution of the expression of endogenous proteins.

5. Why weren’ t lentiviral pseudoptypes with SARS-CoV-2 spike, ideally corresponding to different VOCs not included?

6. The second phenotype of IFITMs, which is to lower infectivity of de novo generated particles, seems to be ignored in this study. How do the bat isoforms behave in corresponding assays?

Reviewer #3: (No Response)

**Part III – Minor Issues: Editorial and Data Presentation Modifications**

Reviewer #1: In the introduction and discussion, the authors discuss gene duplication and alternative splicing as mechanisms for evolutionary innovation in response to viruses. It would be worthwhile for the authors to also address how adaptive evolution through positive selection has influenced IFITM functional diversity in Chiropterans.

Line 382-385: This is a major speculation based on the data presented. Only a limited set of viruses were tested (1 influenza and multiple CoVs), and it’s difficult to say that one is more antiviral than the other without a more representative (diverse viral families) set of viruses. It’s possible that each are specialized in their own unique ways. Perhaps consider toning down this language or discussing potential alternative theories.

In Fig 3, the data is normalized based on vector transfected cells, where Fig S3 labels the control as “untransfected.” Which is correct here? Using untransfected cells as a control is not appropriate as the mere act of transfection may influence the infection. A more detailed gating strategy in S3 would be helpful to see how many cells were successfully transfected.

Fig 3 figure/legend:

(B) For clarity, the authors should note that IAV infection was assessed by NP staining. I had to dig into the supplement figure/methods to find this out. MOIs should also be noted.

(C-E) Similarly, that the lentiviral pseudotype vectors encode a luciferase transgene. It’s implied by the readout, but not explicitly stated in the legend.

(G) I’m confused by “infected with serially diluted.” Do you mean cells were infected and then cell supernatants were serially diluted to quantify infection by TCID50? This also comes up in Fig 6C. Please clarify.

Fig 3. Please make the y-axis consistent for all figures. The y-axis for E and G are different than the others.

Fig 6A. Genus/species names should be italicized.

Line 73: “Dengue” should be “dengue.”

Line 837: these plots don’t match up with their descriptions. “B” is VSV infected, and “C” is poly(I:C) treated, but the legend states “poly(I:C)-treated (B) or VSV-infected (D) cells.” Please correct either the figure labeling or legend.

Lines 361-364: Two different points are being made. Please consider splitting the argument into two sentences.

Reviewer #2: 1. Lines 31-32: ‘the unique IFITM repertoire in different species influence their tolerance to viral infections’: tolerance or resistance?

2. Line 66 “dictates” change to ‘dictate’

3. Line 72: ‘IFITM3 is the most well-studied’ change to ‘IFITM3 is the best-studied’

4. Lines 107, 475 and elsewhere: ‘SARS-CoV-1’ change to ‘SARS-CoV’

Reviewer #3: 1. The rationale for the inability to perform absolute quantification of expression in line 171 and Figure 1 is unclear. Could real-time PCR, targeting each specific isoform, facilitate quantification without the need for RNAseq, given the focus on rsIFITM genes?

2. The Methods section requires elaboration. For instance, the origin or source of the RsKT.01 cell line, details on the transfection procedure, and normalization methods need clarification. In Figure 3A, a disparity in the expression levels of IFITMa and IFITMb proteins is observed; does this contribute to the differing antiviral effects between the isoforms? The stability and passage number of the cell lines used prior to infection should also be clarified. In general, the current description lacks sufficient detail for replication.

3. In lines 229 and 295, the authors discuss the distinct antiviral specificities of rsIFITMa and rsIFITMb (and similarly for raIFITM3a and raIFITM3b), yet the presented data do not convincingly demonstrate this specificity. The study primarily shows that isoform a consistently exhibits stronger antiviral activity than isoform b across Figures 3, 4, and 6. Clarification on the definition of specificity versus activity is needed, especially since the study does not establish a correlation between IFITM localization and antiviral specificity (line 343).

4. The use of the TCID50 assay in Figures 3G and 6C raises questions. This assay does not follow the traditional approach of measuring virus produced in the supernatant. The rationale behind not titrating the virus in the supernatant should be explained. Additionally, the representation of TCID50 values typically employs a log scale, making the 60-80% infectivity relative to the vector control appear minimal. Details on the number of biological replicates, stable cell lines, and independent infections conducted would greatly benefit the reader. Additionally, why TCID50 was used instead of IFA to quantify the infectivity?

PLOS authors have the option to publish the peer review history of their article (what does this mean?). If published, this will include your full peer review and any attached files.

Reviewer #1: No

Reviewer #2: No

Reviewer #3: **Yes: **Bin Zhou
---

## [Decision Letter · Decision Letter 1]

22 Oct 2024

Dear Dr. Sloan,

Thank you very much for submitting your revised manuscript "Alternative splicing expands the antiviral IFITM repertoire in Chinese rufous horseshoe bats" for consideration at PLOS Pathogens. As with all papers reviewed by the journal, your manuscript was reviewed by members of the editorial board and by several independent reviewers, including those who commented on the original version and also a new reviewer. The reviewers appreciated your effort and revised manuscript. Based on the reviews, we are likely to accept this manuscript for publication, providing that you modify the manuscript according to the review recommendations.

Sincerely,

Alexander Gorbalenya

Section Editor

PLOS Pathogens

Michael Malim

Editor-in-Chief

PLOS Pathogens

orcid.org/0000-0002-7699-2064

Reviewer Comments (if any, and for reference):

Reviewer's Responses to Questions

**Part I - Summary**

Reviewer #1: I appreciate the effort put in to addressing this reviewers’ comments. I believe the manuscript has been improved by the additional experiments and corrections made throughout. However, I do have one concern that I would like the authors to address.

Reviewer #2: The authors have addressed my concerns satisfactorily.

-They should please add the source and/or information regarding the production of IAV, VSV and HCoV-229E-GFP, and plaque/TCID50 assays.

-lines 485/486: "at a 1:3 molar ratio of plasmid DNA to transfection reagent", this does not make sense because DNA is in ug and Fugene in ul, so it is not a molar ratio.

-throughout the paper: the correct term is "immunostained" (for antibodies), not "stained"

-throughout the paper: "SARS-CoV and MERS-CoV pseudotypes" please change to "SARS-CoV spike and MERS-CoV spike pseudotypes". The pseudotypes are lenti/retroviral based and contain merely the heterologous glycoprotein/spike/HA of indicated viruses.

Reviewer #4: In this study, Mak et al. explore the IFITM repertoire of Chinese rufous horseshoe bats and its antiviral properties against several human pathogenic zoonotic viruses, such as SARS-CoV2, IAV and Nipah virus. Using a range of molecular techniques, they show that this bat species encodes two IFITM isoforms as a result of alternative splicing and show that the two isoforms exhibit distinct cellular localization and different antiviral properties against various pseudoviruses and replication-competent viruses used in the study. After the first round of revisions, the authors have addressed overall well the first set of reviewer questions and comments, and notable improvements in the manuscript and dataset are evident. However, I do have several additional questions or comments on the text and figures.

**Part II – Major Issues: Key Experiments Required for Acceptance**

Reviewer #1: In response to this reviewers’ comments, the authors performed additional studies described in Supplementary Figure 5. While rsIFITMa consistently shows a restrictive phenotype, the data presented for rsIFITMb now conflicts with that presented in Fig. 3. See revised manuscript lines 213-227. rsIFITMb is described as inhibitory in Fig. 3B (statistically different from vector transfected cells), yet the percent cells infected are on par with that of the vector control in Fig S5B/C (i.e., rsIFITMb is not inhibitory). These are two contradictory phenotypes that warrant discussion, as it potentially calls into question the modest restrictive phenotypes of rsIFITMb for other viruses in Fig. 3 and reproducibility amongst experimental replicates.

Reviewer #2: (No Response)

Reviewer #4: Lines 111-112: “Constitutive expression of IFNs in bats” is a somewhat incorrect statement, due to how broad the statement itself is regarding “bats”. Constitutive expression of IFN has been shown in only a few bat species so far, with emerging evidence across other species showing no evidence of constitutive IFN expression. Therefore, I’d suggest altering this statement to reflect that constitutive IFN expression isn’t actually a Chiroptera-wide phenomenon. It's a feature of some bat species, but not others.

Lines 561-579 (Replication-competent virus infection) and Supplementary Figure 4: Was a dead cell exclusion dye included in the flow cytometry analysis? There is no mention of a live/dead differentiation, which is a crucial first step in any flow cytometry approach. Dead cells are highly autofluorescent in most channels, so failing to exclude them during gating can significantly distort population percentages and overall end results. Not having performed dead cell exclusion would be a big flaw in the experimental design of the flow cytometry data in Fig. S4. Same comment applies to the gating strategy in Fig. S6.

**Part III – Minor Issues: Editorial and Data Presentation Modifications**

Reviewer #1: (No Response)

Reviewer #2: (No Response)

Reviewer #4: Line 416: Minor vocabulary correction - “with horseshoe bats as an exemplar” should be corrected to “with horseshoe bats as an example”.

Perhaps I’ve missed it, but I didn’t see an explanation in the Methods where the RsKT.01 cell line originated, was gifted or was purchased from.

For all the bioinformatics analysis performed in this study, I strongly suggest that the authors make the code scripts they used available on a repository such as GitHub or any other alternative, in the name of transparency and reproducibility.

Lines 493-499: Please include the dilutions of each antibody.

PLOS authors have the option to publish the peer review history of their article (what does this mean?). If published, this will include your full peer review and any attached files.

Reviewer #1: No

Reviewer #2: No

Reviewer #4: No

Figure Files:

Data Requirements:

Reproducibility:

References:

---

## [Decision Letter · Decision Letter 2]

18 Nov 2024

Dear Sloan,

We are pleased to inform you that your manuscript 'Alternative splicing expands the antiviral IFITM repertoire in Chinese rufous horseshoe bats' has been provisionally accepted for publication in PLOS Pathogens.

Best regards,

Alexander E. Gorbalenya, Ph.D., D.Sci.

Section Editor

PLOS Pathogens

Michael Malim

Editor-in-Chief

PLOS Pathogens

orcid.org/0000-0002-7699-2064

Reviewer Comments (if any, and for reference):

Reviewer's Responses to Questions

**Part I - Summary**

Reviewer #4: The authors seem to have addressed most reviewer questions and concerns adequately.

**Part II – Major Issues: Key Experiments Required for Acceptance**

Reviewer #4: N/A

**Part III – Minor Issues: Editorial and Data Presentation Modifications**

Reviewer #4: N/A

PLOS authors have the option to publish the peer review history of their article (what does this mean?). If published, this will include your full peer review and any attached files.

Reviewer #4: No

---

## [Editor Report · Acceptance letter]

19 Dec 2024

Dear Sloan,

We are delighted to inform you that your manuscript, "Alternative splicing expands the antiviral IFITM repertoire in Chinese rufous horseshoe bats," has been formally accepted for publication in PLOS Pathogens.

Best regards,

Sumita Bhaduri-McIntosh

Editor-in-Chief

PLOS Pathogens

orcid.org/0000-0003-2946-9497

Michael Malim

Editor-in-Chief

PLOS Pathogens

orcid.org/0000-0002-7699-2064